# Quantifying the contribution of transport to Antarctic springtime ozone column variability

Hannah E. Kessenich[1], Annika Seppälä[1], Dan Smale[2], Craig J. Rodger[1], and Mark Weber[3]

[1]Department of Physics, University of Otago, Dunedin, New Zealand
[2]National Institute of Water and Atmospheric Research, Lauder, New Zealand
[3]Institut für Umweltphysik, Universität Bremen FB1, Bremen, Germany

**Correspondence:** Hannah E. Kessenich (hannah.kessenich@otago.ac.nz)

**Abstract.**

Quantifying chemical and dynamical drivers of Antarctic ozone variability remains important as stratospheric chlorine levels gradually reduce and the ozone hole recovers in response. While chemistry dominates the formation of the ozone hole in September, the role of dynamics grows as the spring season progresses. To improve our ability to characterise the dynamical impacts on Antarctic total column ozone (TCO), we use MLS/Aura observations of carbon monoxide to trace the path of an air parcel that originates in the mesosphere and descends into the springtime polar vortex. We define a new metric, the Mesospheric Parcel Altitude (MPA), which measures the altitude of the descending mesospheric air parcel at the end of October. The MPA is highly correlated with October TCO and functions as a diagnostic tool, capturing the dynamical state of the inner-vortex. Based on the MPA, we classify October ozone holes from 2004–2024 into three mesospheric descent types (Strong, Regular, and Weak) and provide a formula to estimate the magnitude of horizontal ozone transport (poleward of 70°S and between 17–27 km) during a given October. A higher MPA (>26.9 km) indicates Weak descent, reduced ozone transport, and a larger, longer-lived ozone hole. A lower MPA (<24.6 km) indicates Strong descent, increased ozone transport, and a smaller, shorter-lived ozone hole. When the MPA is used as a proxy for polar cap TCO, approximately 63% of the observed variance during October is explained by the metric.

## 1 Introduction

The Antarctic ozone hole has been a yearly phenomenon since its discovery in the 1980's (Chubachi, 1985; Farman et al., 1985; WMO, 2022). Complete recovery due to gradually declining stratospheric chlorine levels is projected to occur between 2060–2090 (Dhomse et al., 2018; WMO, 2022; Robertson et al., 2023). Early signs of ozone hole recovery have been identified (Solomon et al., 2016; Stone et al., 2018, 2021; Weber et al., 2022; WMO, 2022; Kessenich et al., 2023; Wang et al., 2025), but many complex dynamical drivers contribute to large interannual variability and can mask recovery signals (Salby et al., 2012; Chipperfield et al., 2017; Stone et al., 2018; Ball et al., 2019; WMO, 2022). In recent years (2020–2022), the Antarctic ozone hole has remained unusually large and deep late into the spring season (Kessenich et al., 2023). Such holes present a greater UV risk to Antarctic plants and animals (Barnes et al., 2022; Cordero et al., 2022; Robinson et al., 2024) and cascade circulation effects across the Southern Hemisphere (SH) (Son et al., 2010; Polvani et al., 2011; Thompson et al., 2011). Enhancements in

polar stratospheric cloud formation and earlier onset of chlorine-driven lower stratospheric ozone depletion following volcanic and wildfire events has been the subject of many studies in the last decade (Solomon et al., 2016; Ivy et al., 2017; Stone et al., 2017; Zhu et al., 2018; Yu et al., 2021; Yook et al., 2022; Solomon et al., 2023; Wohltmann et al., 2024; Fleming et al., 2024; Santee et al., 2024), but these effects are often limited to September and do not explain the full extent of recent ozone hole trends in October (Kessenich et al., 2023).

Focusing on the latter half of the ozone hole season, Kessenich et al. (2023) identified a link between October ozone concentrations and changes in air descending from the mesosphere into the polar vortex. Descent into the polar vortex has been studied since the first satellite observations were made of the chemical composition within the Antarctic ozone hole (see e.g. Russell III et al., 1993; Rosenfield et al., 1994; Schoeberl et al., 1995; Bacmeister et al., 1995; Abrams et al., 1996; Allen et al., 1999, 2000; Kawamoto and Shiotani, 2000). Early studies using observations and modelling found evidence of the presence of mesospheric air descending rapidly and remaining unmixed inside the Antarctic polar vortex each austral winter through the spring (Russell III et al., 1993; Fisher et al., 1993; Eluszkiewicz et al., 1995; Bacmeister et al., 1995; Abrams et al., 1996; Lee et al., 2011), with little disturbance from air entering the vortex from outside the vortex edge until mid-November (Bowman, 1990, 1993; Sutton, 1994; Manney et al., 1994; Eluszkiewicz et al., 1995; Schoeberl et al., 1995; Bacmeister et al., 1995; Abrams et al., 1996). This phenomenon has been detected in both hemispheres, with descending parcels of mesospheric air making up large percentages of inner-vortex air at altitudes as low as 25 km (Fisher et al., 1993; Russell III et al., 1993; Eluszkiewicz et al., 1995; Bacmeister et al., 1995; Abrams et al., 1996; Plumb et al., 2002; Engel et al., 2006; Müller et al., 2007; Funke et al., 2014). Driven by the vertical propagation of both gravity and planetary waves, as governed by the "downward control" principle (Lindzen, 1981; Holton, 1983; Andrews et al., 1987; Haynes et al., 1991; Garcia and Boville, 1994; Eluszkiewicz et al., 1995; Siskind et al., 2007), descent from the mesosphere is so significant that the entire mass of the polar mesosphere has been suggested to flush through the polar vortex each spring (Fisher et al., 1993; Sutton, 1994; Plumb et al., 2002).

In the current work, we will investigate the link between mesospheric descent, vortex dynamics, and patterns in Antarctic ozone using a combination of satellite observations and model simulations covering a period of 21 years. Our focus is on the month of October where the dynamical pathway of the ozone hole often diverges, sometimes leading to a small and shallow ozone hole (as in 2019 (Klekociuk et al., 2021)) or one that remains deep and large late into the spring (as in 2020–2022 (Klekociuk et al., 2022)). We seek to formalize the link identified in Kessenich et al. (2023), concluding that descent from the mesosphere serves as a diagnostic indicator of vortex dynamics and horizontal ozone transport. We present a new metric based on this mechanism that allows for a more accurate attribution of dynamical versus chemical drivers of October TCO variation.

## 2  Data and methods

### 2.1  MLS/Aura observations

We use the Microwave Limb Sounder onboard the Aura satellite (MLS/Aura) for Level 2 Version 5 observations of ozone ($O_3$) (Froidevaux et al., 2008; Schwartz et al., 2020) and carbon monoxide (CO) (Pumphrey et al., 2007; Schwartz, M. et al., 2020)

volume mixing ratios (VMR). MLS/Aura provides high-latitude (82°N–82°S), vertically-resolved observations from mid-2004 onwards.

MLS/Aura $O_3$ observations allow us to reconstruct SH springtime ozone profiles from 2004–2024, extended from the lower stratosphere to the mesosphere (with a recommended vertical range of 261–0.0215 hPa) (Livesey et al., 2022). We use the MLS/Aura $O_3$ observations to form two daily profile time series for further analysis: (1) A high-latitude-core selection, and (2) A polar cap zonal average. For the high-latitude-core dataset (1), we perform an iterative process to select an altitude-dependent approximation of the highest latitude air within the core of the ozone hole. Starting with grid points between 75°S–82°S, we

remove coordinates where $O_3$ VMR exceeds an initial filtering threshold of 5 ppmv (assumed to be influenced by an off-pole shift of the polar vortex). These are replaced with the next lower-latitude coordinates (74°S, 73°S,···, 50°S) while retaining the same total area as covered by the initial grid point selection. When 70% of all grid points poleward of 50°S exceed the threshold of 5 ppmv, the filtering threshold is increased to 5.5 ppmv (and then 6 ppmv and 6.5 ppmv iteratively, as needed). If 70% of grid points exceed the final threshold of 6.5 ppmv, we leave the given level unfiltered and assume the vortex is no

longer intact at this day and altitude. The outcome of these steps is a dataset which holds approximately the same surface area at every vertical level, with data points corresponding to the highest latitudes within the ozone hole. This dataset will be used for visualization purposes only. The polar cap dataset (2) is a simple 65°S–82°S latitude selection which is averaged into a single profile for each day and converted into units of number density (molecules $\mathrm{cm}^{-3}$).

     Observations of CO have been used as a dynamical tracer to examine mesospheric mass transport into the lower meso-

75 sphere and upper stratosphere (Allen et al., 1999, 2000; Lee et al., 2011). The primary source of CO in the mesosphere and thermosphere is the photolysis of $CO_2$, creating large reservoirs of CO in the polar night mesosphere which can then descend downward into the lower mesosphere and stratosphere (Solomon et al., 1985). We use MLS/Aura CO observations (recommended vertical range 215–0.00564 hPa) from 2004–2024 to reconstruct the yearly austral winter/spring mesospheric descent into the polar vortex (Livesey et al., 2022). The MLS/Aura CO observations are analysed to create two separate datasets: (1)

A high-latitude zonally-averaged daily profile time series, and (2) A daily descent-tracking time series. For the high-latitude dataset (1), as stratospheric CO concentrations are very low beyond the edge of the polar vortex, we use a simple high-latitude (75°S–82°S) zonal average of daily CO observations. For the descent-tracking dataset (2), we use MLS/Aura CO observations to track the daily location of the centre of the descending mesospheric parcel. We note that this dataset is not intended to represent the mean descent rate of the overall atmosphere within the polar vortex, as appropriately cautioned against by Ryan

et al. (2018).

     To produce the descent-tracking dataset (2), we seek to utilize a location-finding approach that is robust under possible fluctuations in CO concentration: Other authors have identified impacts to mesospheric CO VMR due to solar-cycle variation in $CO_2$ photolysis (Lee et al., 2013, 2018). Additionally, there are potential implications to CO from solar proton events (SPEs), which are known to produce hydroxyl (OH) in the mesosphere (Solomon et al., 1981). Production of OH during the

90 October-November 2003 major SPEs (Jackman et al., 2008) has been implicated in local reduction of CO (Funke et al., 2011). The signal from concentration fluctuations in mesospheric CO can be expected to propagate into the stratosphere within the descending air parcel. Furthermore, slow chemical loss of CO occurs as the parcel descends into the stratosphere, which must

be accounted for (Minschwaner et al., 2010). To combat these sources of CO fluctuation, we choose a method that uses the spatial structure of CO rather than a concentration threshold value (similar to Lee et al., 2011) to track the descent of the
mesospheric parcel. To keep this method easily replicable, we start with the simple high-latitude CO dataset (averaged from 75°S–82°S) rather than one based on the core of the ozone hole. We first apply 10-day rolling average to the high-latitude CO dataset and look for a distinctly separated peak in the CO VMR profile from the larger mesospheric reservoir above, typically occurring once the parcel has descended below 36 km into the middle stratosphere. We then fit an interpolated curve to the CO profile to extract the altitude of the profile's maximum concentration for the day. This is repeated from the day the parcel
descends below 36 km until the day the peak CO VMR drops below 0.02 ppmv (usually around mid-November).

For the generation of the daily time series from MLS/Aura, if any day includes only outliers or data is missing, rather than applying interpolation, we have opted to fill in the missing day with observations from the day before or day after for best continuity.

## 2.2    SBUV Merged Ozone Data Set

Following the typical period of chlorine-driven ozone depletion in the September lower stratosphere (Santee et al., 2008), October usually marks a pivot point when dynamical effects begin to dominate the evolution of the ozone hole. As cooling in response to polar stratospheric ozone depletion maximizes in October–December (Randel and Wu, 1999), the extent of polar ozone loss in October is an important contributor to circulation variation in the SH summer (Thompson et al., 2011). For the context of this work, we will be focusing on TCO values during the month of October only.

We use the SBUV Merged Ozone Data set (MOD) version 8.7 (Frith et al., 2014) to calculate the average October polar cap TCO from 2004 to 2024 (more information is available at http://acdb-ext.gsfc.nasa.gov/Dataservices/merged/). The 2024 values in the MOD dataset, as sourced from the Ozone Mapping and Profiling Suite Nadir-Profiler instrument (Kramarova, 2024), are preliminary but not expected to change. We average TCO values over the latitude range 65°S–90°S, replicating what is commonly used to track the dynamical response to polar cap ozone loss (Thompson et al., 2011; Zambri et al., 2021). We
note that this latitude range likely samples points outside of the inner polar vortex by October, capturing both the area and depth of the ozone hole in combination with the magnitude of ozone transport into the collar region just outside the vortex.

With the SBUV MOD observations, we use a k-means clustering method (Lloyd, 1982) to classify the years from 2004 to 2024 into three categories based on the average TCO in October. We identify a year as having a Minor ozone hole when October TCO is above 259 Dobson Units (DU), a Moderate ozone hole when October TCO is between 208 and 259 DU, and a Major
ozone hole when October TCO is below 208 DU. The individual years classified into each category are presented in Table 1. Henceforth, we will be using these categorical designations. The combination of Minor and Moderate ozone hole categories can additionally be referred to as Non-Major ozone holes. We note that these categories are not intended as definitions, rather they will purely be used within our current study as a means to understand how features of mesospheric descent vary through the spectrum of ozone hole outcomes. For the classification in Table 1, alternative TCO latitude ranges have been considered
(such as 60°S–90°S, not shown). The years included in each ozone hole category are found to be insensitive to such variation.

**Table 1. October ozone hole classifications.** The SBUV Merged Ozone Data Set (SBUV MOD) October total column ozone (TCO) polar means (65°S–90°S) are shown for each year from 2004–2024. TCO is given in Dobson Units (DU), and each year is classified as having a Minor (Non-Major), Moderate (Non-Major), or Major ozone hole (see the TCO thresholds given in each column).

| SBUV MOD Ozone Hole Classification | | | | | |
|---|---|---|---|---|---|
| Minor (Non-Major) TCO > 259 DU | | Moderate (Non-Major) 259 DU ≥ TCO ≥ 208 DU | | Major TCO < 208 DU | |
| Year | Oct. TCO (DU) | Year | Oct. TCO (DU) | Year | Oct. TCO (DU) |
| 2012 | 271 | 2004 | 252 | 2006 | 191 |
| 2019 | 293 | 2005 | 223 | 2008 | 204 |
| | | 2007 | 244 | 2011 | 183 |
| | | 2009 | 232 | 2015 | 162 |
| | | 2010 | 219 | 2018 | 183 |
| | | 2013 | 252 | 2020 | 181 |
| | | 2014 | 235 | 2021 | 173 |
| | | 2016 | 231 | 2022 | 177 |
| | | 2017 | 250 | | |
| | | 2023 | 215 | | |
| | | 2024 | 236 | | |

## 2.3 Model simulations

To model the approximate dynamical and chemical conditions during SH ozone holes from the MLS/Aura time period, we use the Community Earth System Model, version 2 (CESM2) (Danabasoglu et al., 2020) with the Whole Atmosphere Community Climate Model with D-region ion chemistry (WACCM-D) (Verronen et al., 2016) as our atmospheric component. For our simulations, we use a 0.9° latitude by 1.25° longitude grid with 88 vertical levels spanning from the surface to ~140 km. Our simulations are run with specified dynamics (SD-WACCM) (Gettelman et al., 2019), where model dynamics are matched to observed conditions from the Modern-Era Retrospective analysis for Research and Applications, version 2 (MERRA-2) (Gelaro et al., 2017). We run the simulation from 2005–2018, and we produce the following daily outputs: $O_3$; CO; chemical reaction rates for the depletion of the $O_x$ family (O + $O_3$) by the $ClO_x$-$BrO_x$, $NO_x$, and $HO_x$ families; and the total $O_x$ family production rate. The $O_3$ and CO datasets offer modelled replications of the MLS/Aura datasets described above. As $O_3$ density is at least one order of magnitude larger than atomic oxygen (O) over all altitudes of interest in this work (Brasseur and Solomon, 2005), the outputs for chemical changes in the $O_x$ family are considered a good approximation for changes in $O_3$. The benefit of using loss and production rates for the $O_x$ family as a whole is that rapid cycling within the $O_x$ family due to the Chapman cycle will not dominate results. These model outputs will be used to distinguish areas impacted primarily by chemical loss from areas impacted by transport.

## 2.4 Reanalysis Data

We use the eddy heat flux (EHF) calculated from the ERA5 (Hersbach et al., 2023) monthly reanalysis data set (available at https://www.iup.uni-bremen.de/OREGANO/index_proxy.html) as a proxy for the Brewer Dobson Circulation (BDC) (Fusco and Salby, 1999; Randel et al., 2002; Weber et al., 2011, 2018, 2022). Years with strong winter BDC are linked with strong planetary wave propagation and increased EHF, which collectively work to reduce the strength of the springtime polar vortex (Kawamoto and Shiotani, 2000; Kravchenko et al., 2012). Alternatively, years with weak winter BDC are linked with weak planetary wave propagation and reduced EHF, and hence a stronger springtime polar vortex. The EHF is calculated at 100 hPa and averaged over 45°S-75°S for each month from 2004–2023 (Weber et al., 2018). We use the accumulated SH EHF from March–October to represent the total yearly influx of vertically propagating planetary waves into the polar stratosphere (from the time the SH winter polar vortex first forms until the mid-spring). The accumulated EHF term is negative for the SH, so we take the magnitude of the EHF ($|EHF|$) for a hemisphere-nonspecific analysis.

To assess the yearly strength of the SH polar vortex, we use the instantaneous 3-dimensional 6-hourly data collection (Global Modeling and Assimilation Office (GMAO), 2015) from the MERRA-2 reanalysis product (Gelaro et al., 2017). This collection has 42 vertical pressure levels and has global coverage with a horizontal resolution of $0.5° \times 0.625°$. We use the zonal wind (U) data field, which we average into a single vertical profile across 45°S–75°S for each day during September and October from 2004–2024. From this point, we use a maximum-based approach to identify the "peak" zonal wind strength and altitude, similar to that of the MLS/Aura CO descent-tracking dataset. In each day's zonal wind profile, we first find the interpolated maximum of zonal wind strength ($\mathrm{m\,s^{-1}}$). We then find the corresponding altitude ($\mathrm{km}$) of the maximum. The results for each day across September and October are averaged to produce estimates of the peak zonal wind (1) strength ($\mathrm{m\,s^{-1}}$) and (2) altitude ($\mathrm{km}$) for each year.

## 3 Results

### 3.1 Annual descent of mesospheric air inside the Antarctic polar vortex

In Figure 1a–d, we present daily high latitude (75°S–82°S) MLS/Aura CO observations from April through November for two Moderate ozone hole years (panel a: 2005, panel c: 2010) and two Major ozone hole years (panel b: 2006, panel d: 2011). Each panel in Fig. 1a–d shows a large reservoir of CO that begins descending from the mesosphere (above 0.1 hPa) in April. By late-August, each panel shows a parcel of air reaching the middle stratosphere ($<36$ km, $\sim4.9$ hPa) and separating from the reservoir above. This general seasonal pattern is present in all years of MLS/Aura observations from 2004–2024 (see Appendix Figure A1) and confirms the mesospheric origin of the CO-rich air inside the springtime polar vortex. From here on, we will refer to this high latitude CO-rich air as a descending mesospheric air parcel.

Figure 1e-h shows the corresponding daily high-latitude-core MLS/Aura $O_3$ observations for the same years as panels a–d (2005, 2006, 2010, and 2011). Overlaid on the ozone observations, we have added blue circles at 15-day intervals tracking the mesospheric air parcel through to early November (see Methods). The first plotted marker represents the day when the

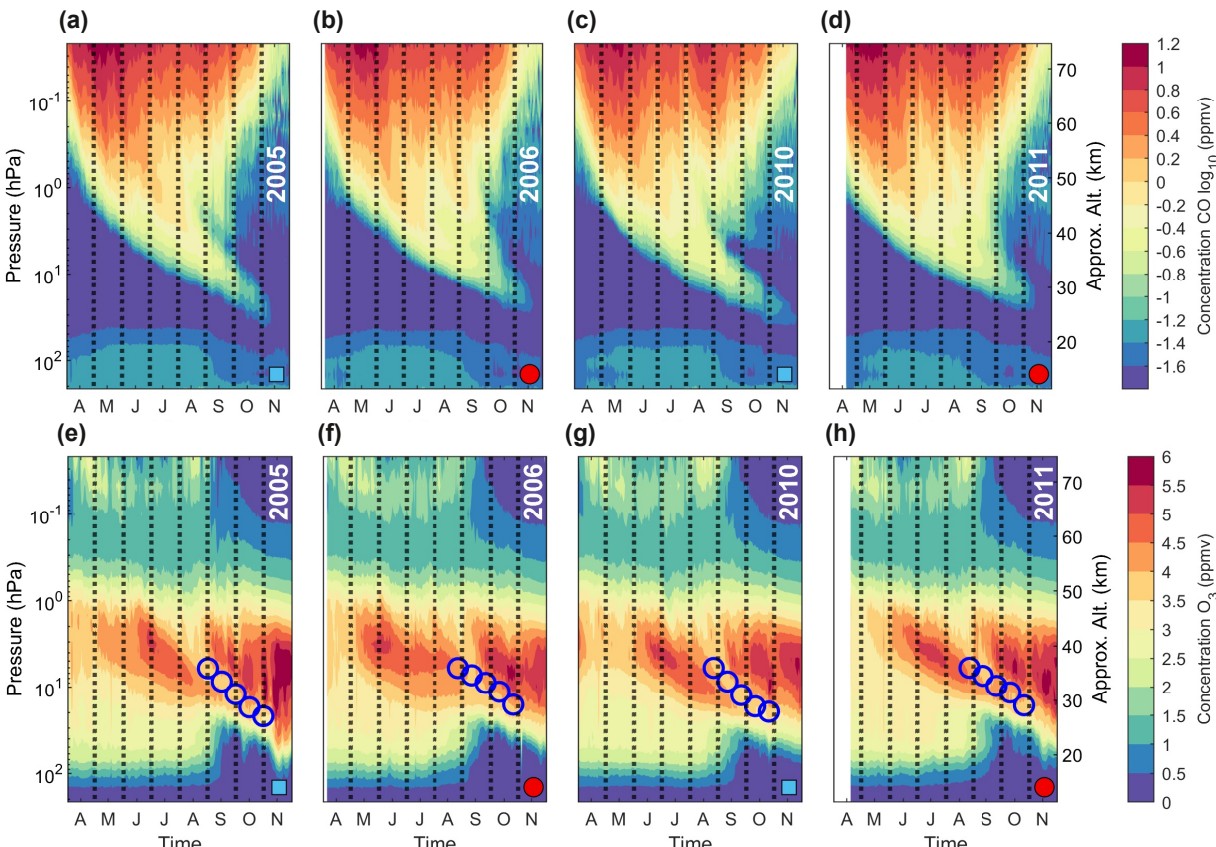

**Figure 1. Sample of MLS/Aura carbon monoxide (CO) and ozone ($O_3$) observations representing two Moderate and two Major ozone hole years.** Daily observations from April to November of **(a)–(d)** zonally averaged high-latitude (75°S–82°S) CO VMR profiles (shown with a logarithmic scale), and **(e)–(h)** high-latitude-core $O_3$ VMR profiles overlaid with blue circles at the altitude of the corresponding day's CO profile maximum below 36 km. Panels are labelled with their corresponding year. Moderate ozone hole years are labelled with cyan squares and Major ozone hole years are labelled with red circles. Months on the $x$-axis are indicated with their corresponding first letter and are separated by black dashed lines.

mesospheric air parcel first descends below 36 km, as this altitude represents a boundary where the air has separated from its CO reservoir above and enters the middle stratosphere (as can be seen in panels a–d). These markers emphasize the distinct

low-$O_3$ signature that coincides with the path of the descending mesospheric parcel. This general pattern is present in all years of MLS/Aura coverage (see Appendix Figure A2) and occurs regardless of a given year's distinction as a Major or Non-Major ozone hole. Additional viewpoints of the mesospheric air parcel during 2005, 2006, 2010, and 2011 are provided in Appendix Figure A3. The maps in Fig. A3 show a pattern of CO-rich air filling the same large areal footprint as the $O_3$-poor signatures, often extending to 60°S.

## 3.2 Characterizing mesospheric descent year-to-year

We now investigate various metrics of the descending mesospheric air parcel to determine which features are linked to larger and deeper October ozone holes (as measured by the October TCO). The metrics considered, all based on annual observations and presented in the corresponding panels of Figure 2, are:

(a) The average daily maximum in CO concentration during October, calculated using MLS/Aura high-latitude CO. We find the magnitude of the CO VMR profile peak for each day in October and convert to units of number density ($10^{12}$ molecules cm$^{-3}$) using the corresponding altitude of the peak. We then average the CO number density results for the month of October.

(b) The day of year when the first point in the CO descent-tracking dataset (see Methods) descends below 36 km in altitude, as this represents when the mesospheric parcel first enters the middle stratosphere and is distinctly detached from its reservoir above (as visualised in Fig. 1).

(c) The descent rate of the mesospheric air parcel (km day$^{-1}$), found using the CO descent-tracking dataset from the day the parcel first descends below 36 km through to November 5th. The November 5th datapoint effectively corresponds to the average CO location at the end of October 31st (averaged over Oct. 27th–Nov. 5th, due to the 10-day rolling average applied). We fit a line to the series, with 36 km fixed as the y-intercept and the slope representing the vertical distance the parcel moves each day.

(d) The final altitude of the mesospheric air parcel at the end of October (in km), represented by the CO descent-tracking data point from November 5th (as above, corresponding to the mean from Oct. 27th–Nov. 5th). This altitude as measured at the end of October will be formally termed the Mesospheric Parcel Altitude, or MPA.

Fig. 2 shows the above metrics on the $x$-axes of the individual panels, with October TCO (as given in Table 1) on the $y$-axes. For each panel, we perform a linear fit between the given metric and October TCO for each year (2004–2024) and report the corresponding R$^2$ value. The markers for each year are coloured to represent each category of ozone hole: Green = Minor, Blue = Moderate, and Red = Major.

Of the results in Fig. 2, two metrics show a strong linear relationship with October TCO: the average descent rate, as shown in panel c (R$^2 = 0.73$), and the Mesospheric Parcel Altitude (MPA), as shown in panel d (R$^2 = 0.86$). Both are calculated after the mesospheric air parcel has spent prolonged time in the stratospheric polar vortex, indicating that the interaction between vertical descent and dynamics within the inner-vortex is important for explaining ozone values. Alternatively, the magnitude of the CO signal (panel a) and the day the parcel first arrives into the middle stratosphere (panel b) are likely primarily influenced by the early-spring strength of gravity wave breaking in the mesosphere (Andrews et al., 1987; Haynes et al., 1991). The lack of correlation between these two metrics and Oct. TCO (particularly in panel b with an R$^2$ of 0.006) indicates that Major ozone holes are unlikely to be significantly driven by early-spring changes in gravity wave driving.

Focusing now on the parcel descent rate and MPA, we separate the daily CO tracking data for the years 2004–2024 into their corresponding ozone hole categories to find the average of these two metrics across the individual years within each category.

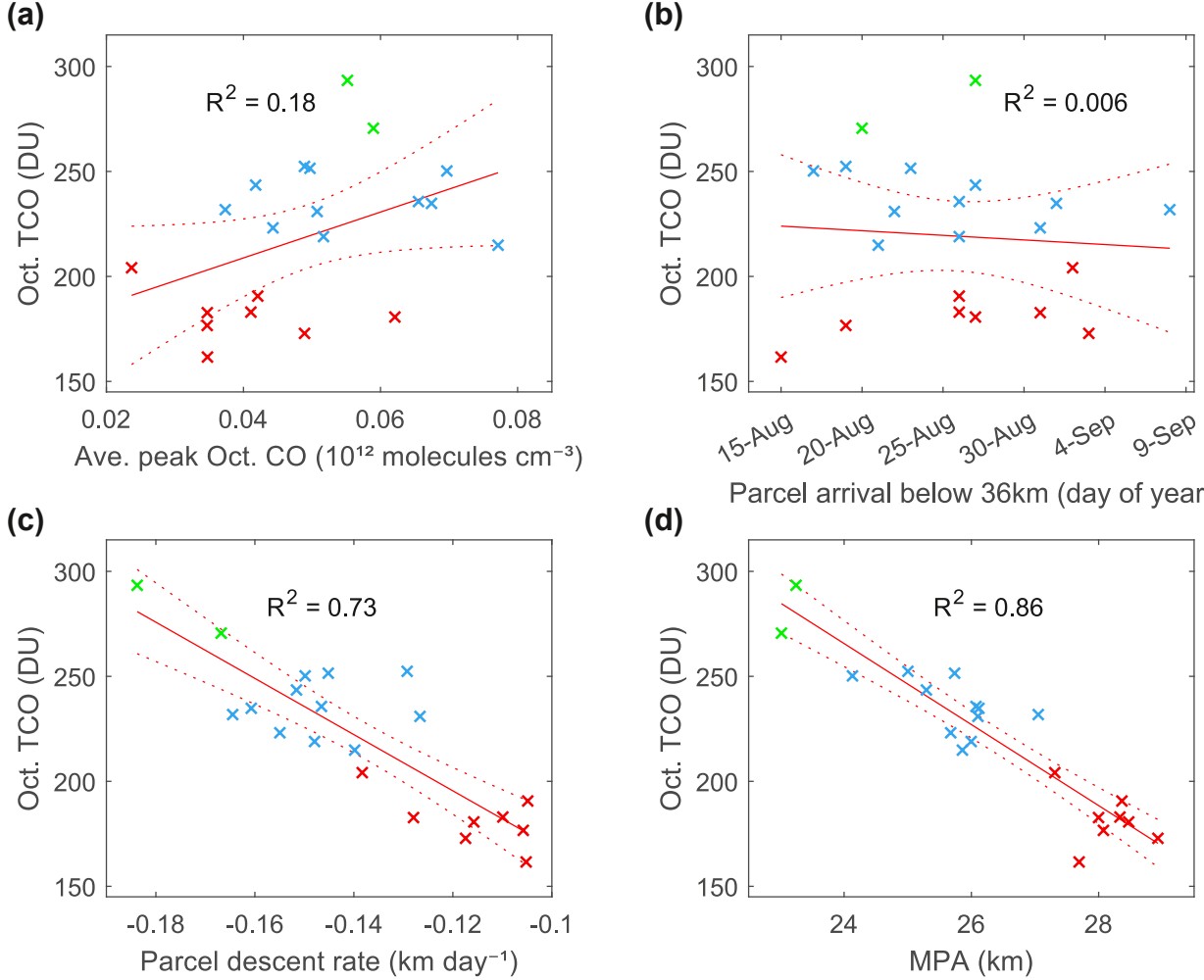

**Figure 2. The relationship between October ozone hole outcomes and various mesospheric descent metrics.** Ozone hole outcomes are indicated by SBUV MOD October average polar (65°S-90°S) total column ozone (TCO) (see Table 1) on the $y$-axis, with $x$-axis representing: **(a)** The average October peak carbon monoxide (CO) value ($10^{12}$ molecules cm$^{-3}$); **(b)** The day of year when the mesospheric parcel first arrives below 36 km (to the middle stratosphere); **(c)** The descent rate of the mesospheric parcel (km day$^{-1}$); and **(d)** The altitude of the mesospheric parcel at the end of October, termed the Mesospheric Parcel Altitude (MPA) (km) (data averaged from Oct. 27th–Nov. 5th). All metrics are calculated from high-latitude (75°S-82°S) MLS/Aura CO data for 2004–2024. A linear fit between October TCO and each metric is performed, with the result and its 95% confidence bounds shown with red lines and the corresponding $R^2$ value presented in each panel. Points for each year are colour-coded according to the categories in Table 1: Green = Minor, Blue = Moderate, and Red = Major ozone hole years.

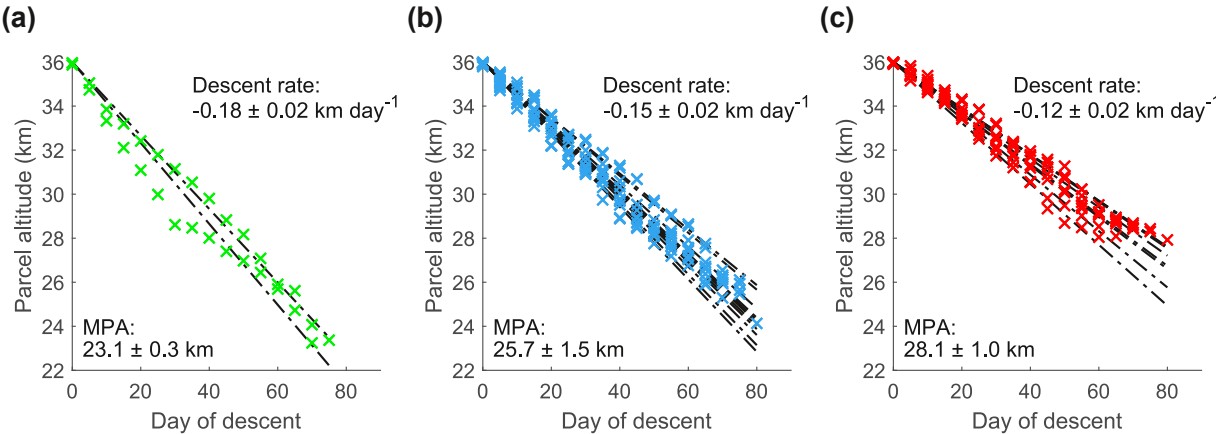

**Figure 3. Average descent rates (below 36 km altitude) and Mesospheric Parcel Altitude (MPA) for each ozone hole category.** Results are shown separately for years with **(a)** Minor ozone holes, **(b)** Moderate ozone holes, and **(c)** Major ozone holes, as listed in Table 1. Data points are shown from the day each year when the centre of the descending mesospheric parcel, given by MLS/Aura high-latitude (75°S–82°S) carbon monoxide (CO) observations, descends below an altitude of 36 km until the end of October. Linear fits to each year's data points are shown with black dash-dot lines. We report the average of the individual group member's slopes (when fitted separately) as the 'Descent rate' and the average location of the mesospheric air parcel at the end of October 31st (in km) for each group as the 'MPA' within each panel. 95% confidence intervals are reported on each metric.

These are presented in Figure 3, with the average descent rate and MPA quoted with 95% confidence intervals and the location of the mesospheric parcel plotted at 5-day intervals for best clarity. The descent rate results across the three categories produce
a three tiered structure, with the fastest descent occurring during Minor ozone holes and the slowest descent occurring during Major ozone holes. The MPA during Major ozone holes remains markedly higher when contrasted to the other two categories, ~28 km for Major ozone holes compared to 23–26 km during Non-Major years.

### 3.3    The mesospheric imprint on ozone profiles

As the parcel of mesospheric air descends into the polar vortex each year, it leaves behind an imprint on $O_3$ profiles (as shown
by the consistent pattern of $O_3$-poor air along the trajectory of the CO-rich parcel in Fig. 1). To formalize our assessment of this signature, we calculate the day-to-day change (i.e. a 'daily delta') of the high-latitude-core MLS/Aura $O_3$ observations. Given the slower and shallower mesospheric descent found during Major ozone hole years (Figures 2–3), we first separate the observations into Major and Non-Major ozone hole years and average the daily profiles within each corresponding category. While already using the high-latitude-core $O_3$ dataset, we then further mitigate any influence from early vortex breakdown at
high-latitudes by excluding data points that exceed 5 ppmv after averaging. We then apply a 10-day rolling average to the data to smooth any abrupt day-to-day VMR fluctuations. Finally, we calculate the daily change in $O_3$ across each altitude using the smoothed data. The results are presented in Figure 4, with Major years shown in panel a and Non-Major years in panel b.

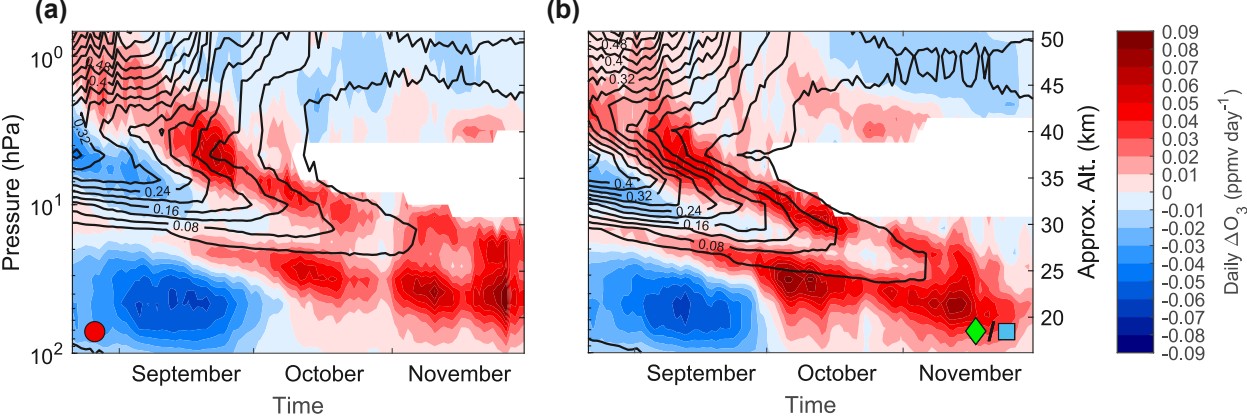

**Figure 4. The imprint of descending mesospheric air on MLS/Aura ozone** ($O_3$) **VMR profiles from 2004–2024.** Day-to-day change ('daily delta') of the high-latitude-core $O_3$ profiles (colours represent the change in $\mathrm{ppmv\,day^{-1}}$), separated into **(a)** Major ozone hole years (labelled with a red circle) and **(b)** Non-Major ozone hole years (labelled with a green diamond and cyan square). Overlaid solid black contours illustrate the average location of the MLS/Aura carbon monoxide (CO) within each category (contour lines shown at 0.04 ppmv increments). White regions indicate excluded data due to average VMR values over 5 ppmv. See text for details of the calculation.

Blue shading depicts regions where $O_3$ at that given altitude is actively reducing, while red shading depicts altitudes where $O_3$ is increasing/rebounding back to baseline concentrations. The white areas correspond to the above mentioned excluded data
points. Both panels include an overlay of black contours to illustrate where the average CO descent from MLS/Aura is found for each category (shown at 0.04 ppmv intervals).

The daily delta results presented in Fig. 4 indicate a general pattern of increasing $O_3$ along the upper and lower altitude boundaries of the descending mesospheric parcel, consistent for all years regardless of ozone hole category. We hypothesise that this pattern could be the result of either (1) chemical production of $O_3$, or (2) transport of $O_3$. The former is unlikely, as
we do not expect the rate of photolysis at these high-latitudes to be sufficient for substantial $O_3$ increase to occur at this time. However, as this question can not be confirmed from the observations alone, we will use model results in the following section to investigate further.

### 3.4 Disentangling chemistry from transport

To test whether the ozone resurgence around the descending parcel shown in Fig. 4 is a transport effect or the result of
chemical production, we use vertically-resolved CESM2 simulations to assess $O_x$ (O + $O_3$) chemical production and loss rates. To provide context for the model results, we first plot MLS/Aura high-latitude-core $O_3$ daily deltas for one Moderate ozone hole year (2005) and one Major ozone hole year (2006) from August 21st through November with black contour lines only (similar to Fig. 4, but now for individual years). We then add the model-calculated daily high-latitude (75°S–82°S) $O_x$ chemical production and loss rates from various contributors as coloured contour lines and shading in separate panels: overall

production of $O_x$ = solid green line, loss due to $ClO_x$ and $BrO_x$ = dashed red line, loss due to $NO_x$ = dashed yellow line, and loss due to $HO_x$ = dashed magenta line. For all chemical production (loss) terms, the contour line corresponds to the same positive (negative) daily VMR change of $0.02 \, \mathrm{ppmv \, day^{-1}}$, with shading showing where each term exceeds this threshold. The results are shown in Figure 5, with 2005 in panel a and 2006 in panel b.

In both example years of Fig. 5, lower stratospheric $O_x$ loss by $ClO_x$ & $BrO_x$ (red) has ceased by early October, with the
250 2006 boundary extending slightly later. However, these regions are similar enough (as are the daily $O_3$ delta values shown in dotted contours beneath) to suggest that Major and Non-major ozone holes during October are unlikely to be driven by differences in September chemistry. Also in both example years of Fig. 5, the lower boundary of the $O_x$ production contour (green line) is nearly entirely overlapped to lower altitudes by the $NO_x$ loss boundary (yellow line) and is also closely matched with both the $HO_x$ (magenta) and the uppermost $ClO_x$ & $BrO_x$ (red) boundaries. This indicates that during October, chemical
$O_x$ production is likely negligible, or fully outweighed by chemical loss, from approximately 27 km and below. Meanwhile, the MLS/Aura $O_3$ positive daily delta is upwards of $0.04 \, \mathrm{ppmv \, day^{-1}}$ between 20–27 km from October onward. While we only show the model results for two years in Fig. 5, this pattern is consistent across the full range of modelled years (2005–2018) (not shown). These results indicate that chemical $O_x$ production is unlikely to explain the resurgence of ozone beneath the descending mesospheric parcel, pointing instead towards horizontal transport.

To form a generalized picture of the "corridor" of horizontal $O_3$ transport that coincides with mesospheric descent over the pole, we subtract the simulated net $O_x$ chemical change (total production minus total loss) from the high-latitude (75°S–82°S) CESM2 simulated $O_3$ daily change. For best visualization of the low-concentration, high-density $O_3$ impact in the region beneath the descending mesospheric parcel, we present these results in number density ($\mathrm{molecules \, cm^{-3}}$). Model data from 2005–2018 is separated into Major and Non-Major ozone hole years (see Table 1). To provide a generalised view of
the transport magnitude, we average results across the years and apply smoothing (a 20-day rolling average). This estimated transport is presented with grey shading in Figure 6a for Major years and in Figure 6b for Non-Major years. We note that all shading represents positive transport values: no points have a negative transport magnitude exceeding the minimum threshold for shading ($-0.025 \times 10^{12} \, \mathrm{molecules \, cm^{-3} \, day^{-1}}$). We add a blue contour line and shading at the location of high-latitude CESM2 CO (also averaged over the included years). To determine the latitudinal extent of the transport pattern, we then
zonally average the modelled net $O_x$ chemical change from 50°S–82°S for the month of October. To pair with this, we find the total amount of modelled $O_3$ change during October by zonally averaging the CESM2 $O_3$ data from 50°S–82°S for October 31st and 1st and finding the difference between the two days. The modelled chemical change in $O_x$ is subtracted from the total $O_3$ change to estimate the total amount of transported $O_3$ during October. As in panels a–b, we separate into Major and Non-Major modelled years. The estimated $O_3$ transport is presented with the grey shading in Figure 6c for Major years and in
Figure 6d for Non-Major years. We again note that all shading represents positive transport values: no points have a negative transport magnitude exceeding the minimum threshold for shading ($-1 \times 10^{12} \, \mathrm{molecules \, cm^{-3}}$). We add a cyan contour line and shading where the total modelled change in $O_3$ during October is $\geq 1 \times 10^{12} \, \mathrm{molecules \, cm^{-3}}$. Areas covered by the cyan shading indicate regions where transport outmatches chemical depletion of $O_3$, and areas not covered by the cyan shading

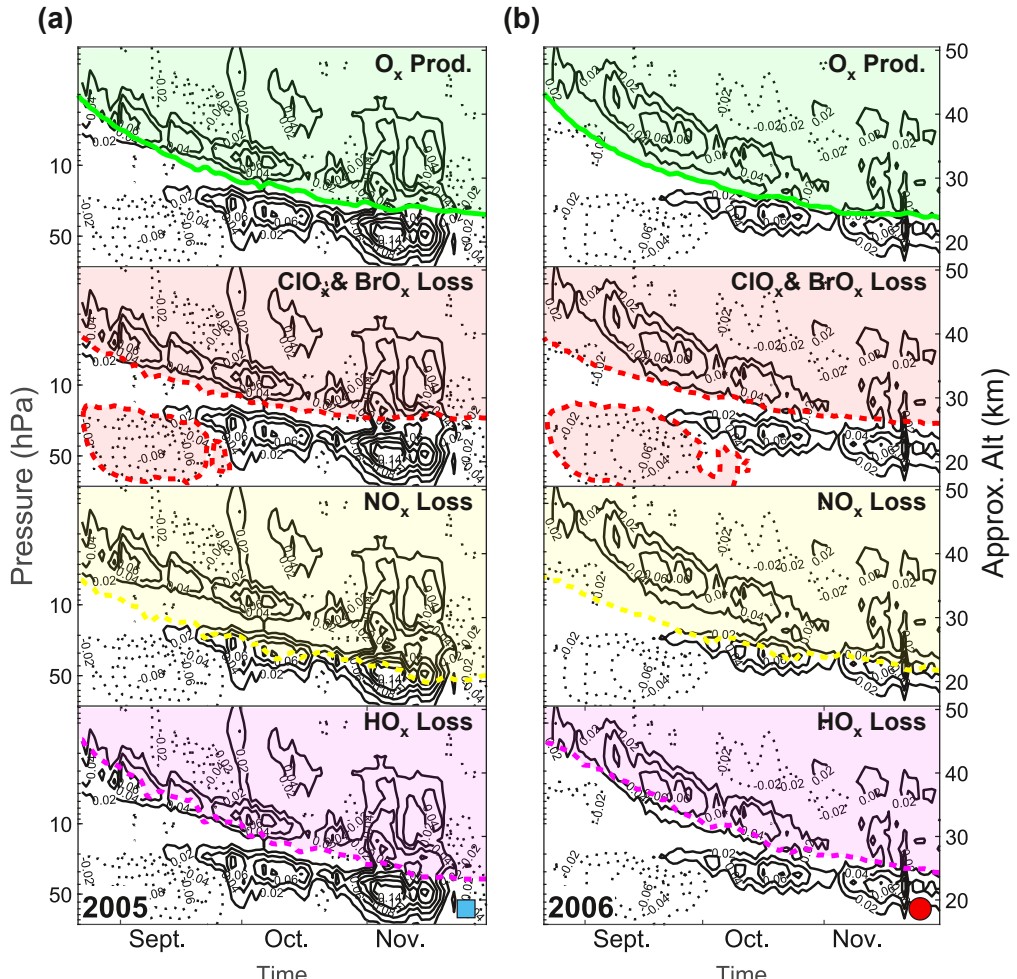

**Figure 5. Odd oxygen family ($O_x$) chemistry contrasted to daily ozone ($O_3$) changes.** Results are shown for **(a)** 2005 (Moderate ozone hole year, labelled with a cyan square) and **(b)** 2006 (Major ozone hole year, labelled with a red circle). The daily change in MLS/Aura high-latitude-core $O_3$ VMR from August 21st through November is shown with black contour lines at increments of $0.02$ ppmv day$^{-1}$ (positive with solid lines, and negative with dotted lines). Overlaid are CESM2 simulated high-latitude (75°S–82°S) $O_x$ daily VMR changes, with contributors separated into: $O_x$ production (green, solid), loss due to $ClO_x$ and $BrO_x$ (red, dashed), loss due to $NO_x$ (yellow, dashed), and loss due to $HO_x$ (magenta, dashed). The contour line and shading for the $O_x$ production term is placed where $O_x$ production $\geq$ $0.02$ ppmv day$^{-1}$. For the three loss terms, the contour line and shading is placed where the magnitude of $O_x$ loss $\geq 0.02$ ppmv day$^{-1}$.

indicate regions where transport and chemical effects are essentially balanced. As in panels a–b, the results are averaged across
the included years to provide a generalized picture of the role of transport.

     The transport patterns between $\sim$27–40 km in Fig. 6c–d have a pole-centric structure. The majority of this structure is the result of modelled $NO_x$-driven $O_x$ loss within the descending air parcel, occurring primarily above 27 km (detectable by

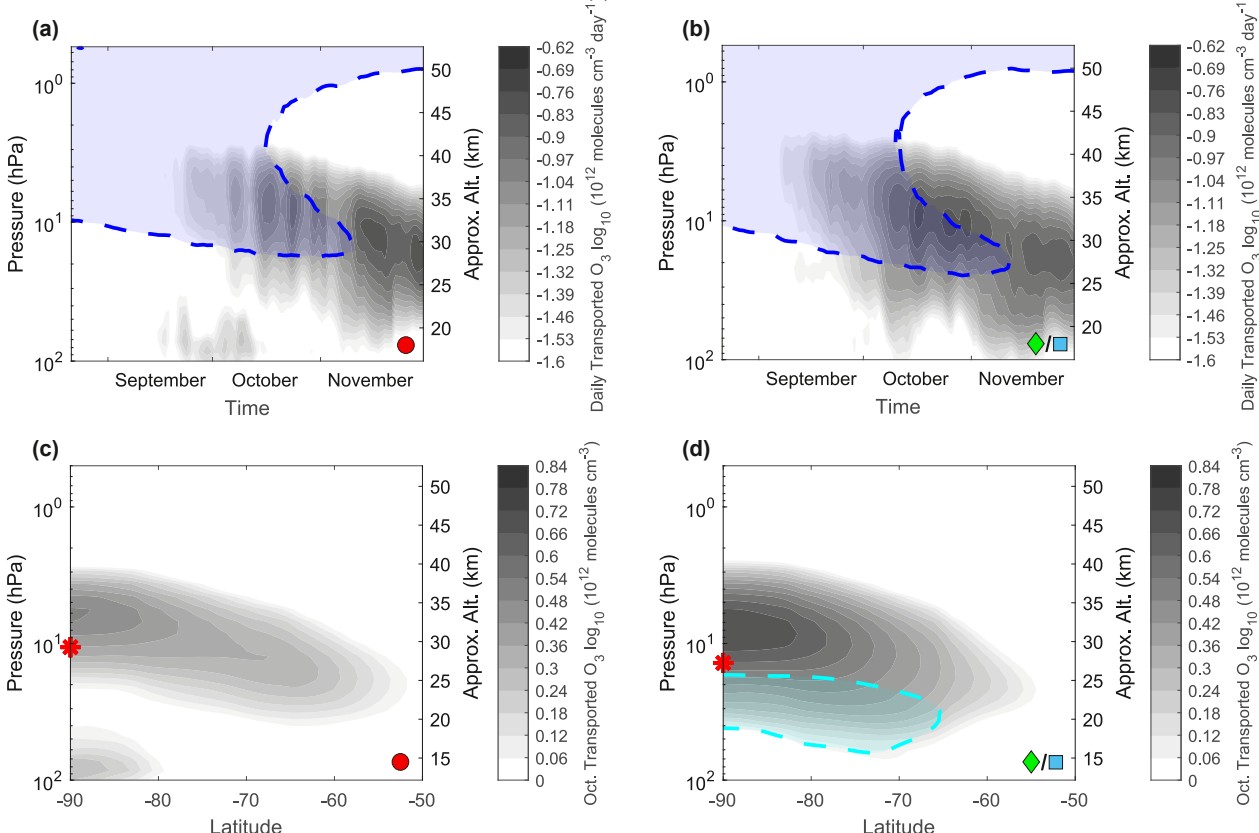

**Figure 6. Model estimated ozone** ($O_3$) **transport over the pole**. Approximations of daily transport of $O_3$ for **(a)** Major (red circle) and **(b)** Non-Major (green diamond and cyan square) ozone hole composites, shown in units of daily number density change ($molecules\,cm^{-3}\,day^{-1}$). $O_3$ transport estimated by using CESM2 simulated high-latitude (75°S–82°S) daily $O_3$ deltas from August 21st through November and subtracting the daily net $O_x$ chemical production ($O_x$ production - $O_x$ loss). The remaining positive $O_3$ changes (plotted in grey shading) are likely dominated by transport effects. The CESM2 high-latitude carbon monoxide (CO) daily $\geq 0.04$ ppmv VMR contour is shown in blue. Approximations of October zonal mean total transport of $O_3$ between 50°S and 90°S for **(c)** Major (red circle) and **(d)** Non-Major (green diamond and cyan square) ozone hole composites. The estimate is established by subtracting the October sum of the daily CESM2 $O_x$ chemical production (as in panels a–b) from the total modelled $O_3$ change from the start to the end of October. The cyan contour and shading illustrate where the total modelled $O_3$ change is $\geq 1 \times 10^{12}$ $molecules\,cm^{-3}$. We add a red asterisk at the altitude of the average modelled Mesospheric Parcel Altitude. Results for model years 2005–2018 are included.

deconstructing $O_x$ loss into its constituent parts as in Fig. 5, not shown). Given that total modelled $O_3$ undergoes negligible changes in this region (no cyan contour), estimated transport values compensate for the severity of modelled $O_x$ loss. A further

investigation into the magnitude of modelled $NO_x$-driven $O_x$ loss within the descending mesospheric parcel (e.g., Szeląg et al., 2022) is warranted, but outside the scope of this study. Here, we focus on transport that occurs beneath the MPA (red asterisk)

between 17 and 27 km, where an additional region of transport is present during Non-Major ozone hole years (panel d). This pattern overlaps with an increase in total modelled $O_3$ (cyan contour, panel d). A similar pattern can be seen in panels a and b, with enhanced transport occurring below the descending mesospheric parcel (blue CO contour) during Non-Major ozone hole years (panel b). For every modelled year, we find that (1) $O_x$ loss is greater than $O_x$ production and (2) transport is positive, at all altitudes below 43 km and poleward of 70°S. From this, we can exclude the likelihood of ozone resupply from higher altitudes and conclude that the pattern between 17–27 km is most likely the result of horizontal ozone transport. Given the latitudinal extent of this pattern (ranging from 65°S to 90°S), it most likely represents ozone resupply that extends into the polar vortex. There is also a third, small-magnitude region of transport seen in Fig. 6a and c below 17 km, the origin of which is beyond the scope of this study.

### 3.5 Mesospheric Parcel Altitude as an observable proxy for ozone transport

The results presented in Figs. 5–6 suggest that there is a crucial difference in poleward ozone transport from ~17–27 km during Major and Non-Major ozone hole years. It has been well established that a strong polar vortex presents an effective barrier to horizontal transport towards the pole (Bowman, 1990, 1993; Sutton, 1994; Manney et al., 1994; Eluszkiewicz et al., 1995; Schoeberl et al., 1995; Bacmeister et al., 1995; Abrams et al., 1996), suggesting that this difference is likely a result of stronger (weaker) vortex conditions during Major (Non-Major) ozone hole years. In Fig. 2, we also identify that the MPA, a metric calculated after the descending mesospheric parcel has spent considerable time within the polar vortex, has the highest correlation with October TCO of all the metrics considered. The yearly strength of the springtime polar vortex is linked with the magnitude of the accumulated EHF, as planetary waves propagate into the stratosphere and act to weaken the zonal winds (Kawamoto and Shiotani, 2000). We therefore hypothesise that the MPA fundamentally reflects the strength of the accumulated EHF and the resulting inner-vortex dynamical conditions, which then control the magnitude of horizontal $O_3$ transport towards the pole. Model simulations, ERA5 accumulated EHF, and MERRA-2 zonal wind fields are used to test these links to the MPA. We calculate the correlation coefficients between each of the following metrics: (i) Modelled CESM2 MPA (km); (ii) Model-estimated horizontal $O_3$ transport (DU), calculated between 70°S–90°S and from 17–27 km; (iii) The average strength of the peak MERRA-2 zonal wind (U, $m\,s^{-1}$), calculated between 45°S–75°S from the start of September to the end of October (see Methods); (iv) The average altitude of the peak MERRA-2 zonal wind (U, $m\,s^{-1}$), calculated between 45°S–75°S from the start of September to the end of October (see Methods); (v) the magnitude of the ERA5 SH EHF (|EHF|, $K\,m\,s^{-1}$), calculated at 100 hPa, averaged over 45°S–75°S, and accumulated from March–October (see Methods); and (vi) Observed MLS/Aura MPA (km). The MPA metrics are calculated from high-latitude (75°S-82°S) CO data from CESM2 and MLS/Aura respectively. Modelled metrics are calculated from 2005–2018, the MLS/Aura and MERRA-2 metrics are calculated from 2004–2024, and the ERA5 metric is calculated from 2004–2023. The latitude range 70°S–90°S is selected for the model-estimated horizontal $O_3$ transport calculation, as this was found to produce the best results during the correlation analysis. We present the resulting correlation coefficients ($r$) as a matrix in Figure 7a.

Fig. 7a shows a high negative correlation ($r = -0.91$) between modelled MPA and model-estimated $O_3$ transport, confirming that the relationship between high (low) MPA and reduced (enhanced) transport is captured in CESM2. We also find a high

positive correlation ($r = 0.93$) between the modelled and MLS/Aura MPA, although we note that the modelled MPA values are consistently biased to higher altitudes (see Fig. A4e). The MLS/Aura MPA can be directly linked with modelled transport ($r = -0.90$), which will be discussed further below. The EHF shows strong links with both zonal wind metrics ($r = -0.87$ and $r = -0.85$), and both zonal wind metrics show strong links with the MLS/Aura MPA ($r = 0.92$ and $r = 0.9$). The signs of the correlations indicate the following: during years with stronger accumulated EHF, the polar vortex weakens and descends to lower altitudes during September and October, and the MPA follows suit. The MPA offers a single, observable diagnostic that captures the accumulated inner-vortex conditions at the end of October.

To formalize the link between the MPA and transport, we present the yearly (vi) MLS/Aura MPA (km) and the (ii) model-estimated $O_3$ transport (DU) from 2005–2018 in Figure 7b. Markers are again coloured according to the ozone hole categories from Table 1 (Green = Minor, Blue = Moderate, and Red = Major ozone hole years). We perform a linear fit between the two metrics and add the 95% confidence bounds on the fit with red lines. The corresponding $R^2$ value is reported. The results in Fig. 7b demonstrate that the MPA is highly explanatory ($R^2 = 0.82$) for the model estimated $O_3$ transport. Appendix Figure A4 provides similar outputs as Fig. 7b for each of the remaining metric pairs from Fig. 7a. As a whole, Fig. 7 demonstrates a link between the EHF, the relative dynamical conditions within the inner-vortex, the MLS/Aura MPA, and the magnitude of ozone transported into the polar vortex (poleward of 70°S) between 17–27 km in October.

A diagram is provided in Figure 8 to complete the conceptual picture of the MPA and its links to polar dynamics. Panel a shows the initial descent of mesospheric CO within the polar vortex. Early in the spring, a parcel of mesospheric CO separates from the reservoir above and continues descending into the middle stratosphere (see Fig. 1), as shown in panel b. Panel c depicts the approximate location of the parcel in mid-October. By now, the vortex has weakened, and horizontal $O_3$ transport is occurring along the lower boundary of the parcel (see Fig. 6). Finally, panel d shows the state of the polar vortex, the ozone layer, and the mesospheric parcel on October 31st. The MPA is measured by finding the altitude of the mesospheric parcel at this time. During Major ozone hole years, the vortex winds remain stronger and higher, the MPA is higher, and less $O_3$ transport occurs during October (see Fig. 7). An animated version of Fig. 8 is provided as a Supplement.

## 3.6 Quantifying transport effects on October total column ozone

From the linear fit presented in Fig. 7b, we provide the following equation to calculate the model-estimated horizontal ozone transport during a given year,

$$O_3 \text{ Transport} = a - b \times \text{MPA}, \tag{1}$$

where 'MPA' is the MLS/Aura MPA (km), '$O_3$ Transport' is the CESM2 estimated horizontal ozone transported poleward of 70°S and between 17–27 km in October (DU), and the regression coefficients have the values of $a = 417$ DU and $b = 13.5$ DU km$^{-1}$. We note that this equation is specific for the fit between ozone transport and the *MLS/Aura* MPA: the CESM2 MPA could be used as an alternative to the MLS/Aura MPA only if the linear bias between the two is accounted for appropriately. The MLS/Aura MPA values for 2004–2024 are provided in Table 2, and we add the predicted ozone transport values as calculated with Equation (1). We repeat a k-means clustering analysis (Lloyd, 1982) to reclassify ozone holes now

**(a)**

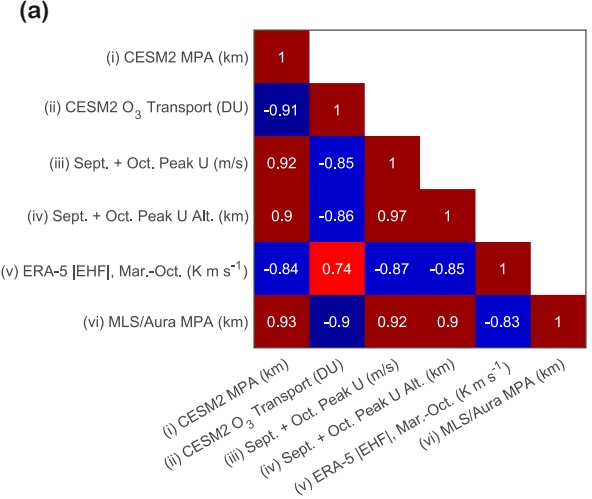

**(b)**

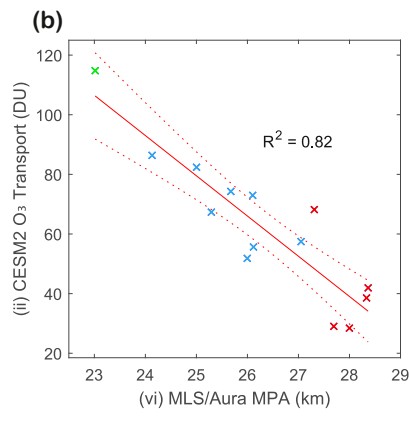

**Figure 7. Mesospheric Parcel Altitude (MPA) linked with modelled horizontal ozone ($O_3$) transport. (a)** Correlation coefficients between (i) modelled CESM2 MPA (km) for 2005–2018, (ii) model-estimated horizontal $O_3$ transport between 70°S–90°S from 17–27 km (DU) for 2005–2018, (iii) the average strength of the peak 45°S–75°S MERRA-2 zonal wind (U) from the start of September to the end of October ($m\,s^{-1}$) for 2004–2024, (iv) the average altitude of the peak 45°S–75°S MERRA-2 zonal wind (U) from the start of September to the end of October (km) for 2004–2024, (v) the magnitude of the ERA5 eddy heat flux (|EHF|) accumulated from March–October ($K\,m\,s^{-1}$) for 2004–2023, and (vi) observed MLS/Aura MPA (km) for 2004–2024. The MPA metrics (i/vi) are calculated from high-latitude (75°S-82°S) carbon monoxide (CO) data. **(b)** (vi) Observed MLS/Aura MPA (km) plotted against (ii) model-estimated horizontal $O_3$ transport between 70°S–90°S from 17–27 km (DU). A linear fit is performed, with the result and its 95% confidence bounds shown with red lines and the corresponding $R^2$ value presented in the panel. Points for each year are colour-coded according to the categories in Table 1: Green = Minor, Blue = Moderate, and Red = Major ozone hole years.

based on the MPA. We identify three distinct groups, which we will refer to as having Strong mesospheric descent, Regular mesospheric descent, or Weak mesospheric descent. We note that 2009, previously identified as a Moderate ozone hole year, now falls inside the Weak mesospheric descent category. While its MPA (27.1 km) places this year within the Weak descent cluster (see the blue marker nearest the cluster of red markers in Fig. 7b), it remains the lowest MPA in the category and thus is a borderline case. The year 2017, also a Moderate ozone hole, now lands in the Strong mesospheric descent category. This year also represents a borderline case, but the dynamics as specified by the MPA (24.1 km) are more closely aligned with the other two Strong descent years (2012 and 2019).

We present in Figure 9a the springtime MLS/Aura $O_3$ composite difference between the Weak descent and Strong/Regular descent ozone hole years (from Table 2), plotted in units of number density ($10^{12}\ molecules\,cm^{-3}$). The latitude range here is extended to include 65°S–82°S to provide an expanded view across the wider polar cap to match our TCO range. Areas with negative (blue) values in Fig. 9a indicate local deficiency of $O_3$ during a Weak descent ozone hole year, due primarily to reduced horizontal transport as indicated by Fig. 6. We add hatching to Fig. 9a where the ozone number density at a given

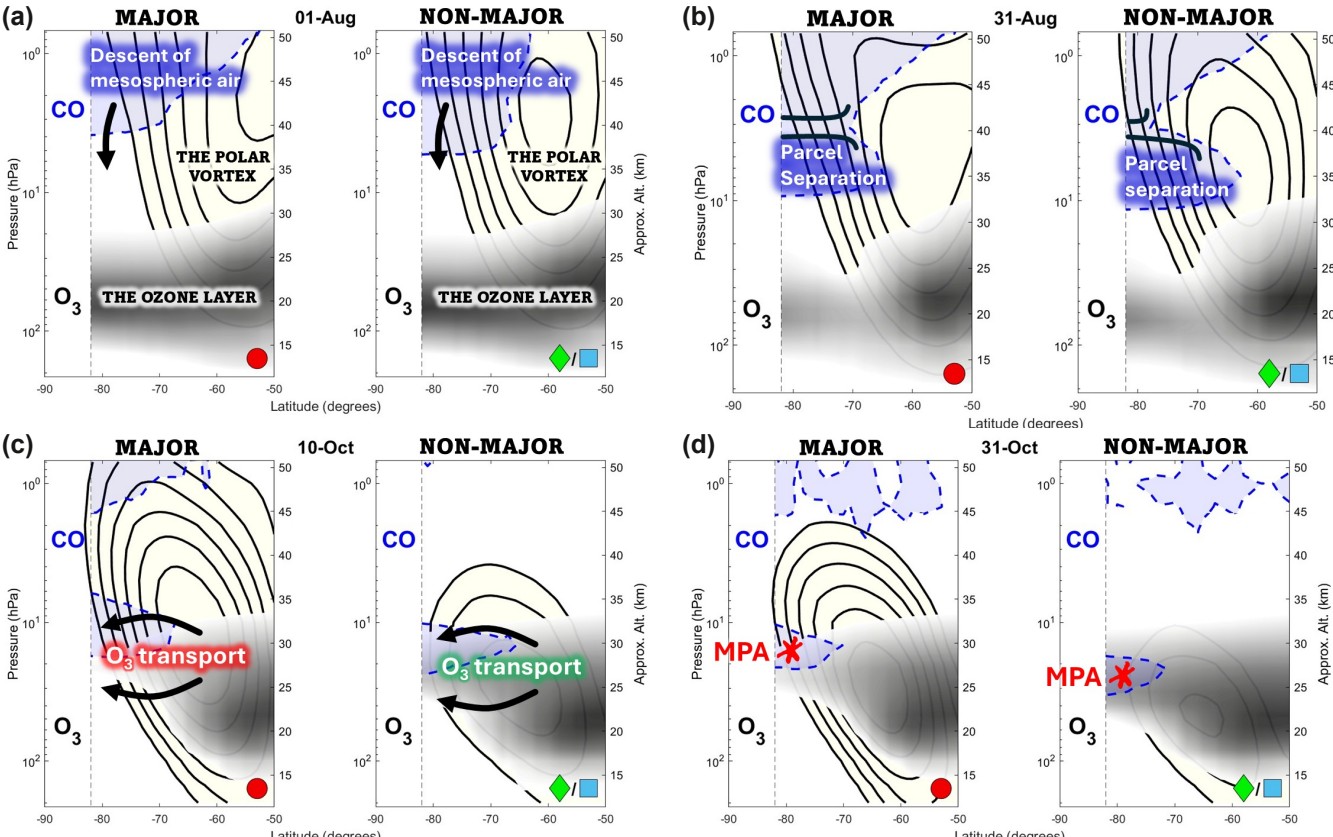

**Figure 8. A conceptual diagram for the Mesospheric Parcel Altitude (MPA) during Major (red circle) and Non-Major (green diamond and cyan square) ozone hole years.** Each panel shows side-by-side frames depicting conditions during Major (left) and Non-major (right) ozone hole years. All frames show latitude on the $x$-axis and altitude on the $y$-axis, with the southern pole at the left. **(a)** Descent of mesospheric carbon monoxide (CO) over the pole has already begun prior to the start of spring. **(b)** During August, a parcel of mesospheric air begins to separate from the reservoir above. **(c)** The mesospheric parcel continues descending, and poleward ozone ($O_3$) transport occurs during September and October. **(d)** The altitude of the MPA is measured at the end of October. By this time, the MPA has captured the dynamical conditions of the inner-vortex, and significant horizontal $O_3$ transport has occurred. During Major ozone hole years, the polar vortex is stronger, descent is slower, poleward $O_3$ transport is reduced, and the MPA is higher.

point correlates highly with the MPA ($|r| \geq 0.7$, significant at the 95% level). While the ozone hole classifications have changed slightly between the TCO categories in Table 1 and the MPA categories in Table 2 (as discussed above), the results from Fig. 9a indicate that Weak mesospheric descent years show a consistent pattern of $O_3$ deficiency through the mid-spring onwards. We then demonstrate the contribution of the MPA metric to year-to-year variation in ozone hole outcomes in Figure 9b. The dashed purple line shows the observed polar cap October TCO values as given in Table 1, with a standard deviation of ~35 DU. We then perform a linear fit between the MPA and the TCO, and we plot the residual TCO in green (relative to the average TCO,

**Table 2. Mesospheric descent ozone hole classifications.** Mesospheric Parcel Altitude (MPA) and model estimated ozone increase due to horizontal transport (DU) from 70°S–90°S and 17–27 km are given from 2004–2024. Years are classified as having Strong, Regular, or Weak mesospheric descent.

| Ozone Hole Mesospheric Descent Classification | | | | | | | | |
|---|---|---|---|---|---|---|---|---|
| Strong Descent MPA < 24.6 km | | | Regular Descent 24.6 km ≤ MPA ≤ 26.9 km | | | Weak Descent MPA > 26.9 km | | |
| Year | MPA | Oct. $O_3$ Transport | Year | MPA | Oct. $O_3$ Transport | Year | MPA | Oct. $O_3$ Transport |
| 2012 | 23.0 km | 106±14 DU | 2004 | 25.7 km | 70±7 DU | 2006 | 28.4 km | 34±10 DU |
| 2017 | 24.1 km | 91±11 DU | 2005 | 25.7 km | 70±7 DU | 2008 | 27.3 km | 48±7 DU |
| 2019 | 23.2 km | 103±14 DU | 2007 | 25.3 km | 76±7 DU | 2009 | 27.1 km | 52±7 DU |
| | | | 2010 | 26.0 km | 66±6 DU | 2011 | 28.3 km | 35±10 DU |
| | | | 2013 | 25.0 km | 79±8 DU | 2015 | 27.7 km | 43±8 DU |
| | | | 2014 | 26.1 km | 64±6 DU | 2018 | 28.0 km | 39±9 DU |
| | | | 2016 | 26.1 km | 65±6 DU | 2020 | 28.5 km | 33±11 DU |
| | | | 2023 | 25.9 km | 68±6 DU | 2021 | 28.9 km | 26±12 DU |
| | | | 2024 | 26.1 km | 65±6 DU | 2022 | 28.1 km | 38±9 DU |

for a best comparison). The green line in Fig. 9b reflects the approximate yearly variation in TCO due to non-transport effects (e.g. chemistry or other dynamical effects not captured by the MPA), with a new standard deviation of ∼13 DU.

## 4 Discussion

As shown in Fig. 1, an air parcel from the mesosphere (as represented by MLS/Aura CO observations) descends into the stratospheric polar vortex each spring. As it descends, the parcel remains intact and traceable, usually reaching the lower stratosphere by mid-October. From MLS/Aura $O_3$ observations within the high-latitude ozone hole, a characteristic "cutout" in the ozone layer can be clearly visualized along the path of the descending mesospheric air parcel through to November.

We explore the connection between ozone hole outcomes in October (as indicated by the polar cap TCO results in Table
1) and characteristics of the CO-rich parcel of descending mesospheric air. When using CO data to track the descent pattern of the air parcel, we find that slower and shallower mesospheric descent is present during Major ozone holes (see the strong linear trends in Fig. 2c–d and the three-tiered dynamical metric structure in Fig. 3). The Minor ozone holes fall at the opposite dynamical extremes to the Major years, with notably faster descent rates and the parcel reaching as low as 23 km in altitude by the end of October. Rapid mesospheric descent during 2019, a known SSW year (Yamazaki et al., 2020; Klekociuk et al.,
2021), fits well within the current understanding of vortex dynamics during SSWs where downwards transport is accelerated following the event (Siskind et al., 2010). The parcel descent rate (Fig. 2c) and MPA (Fig. 2d) are both calculated after the

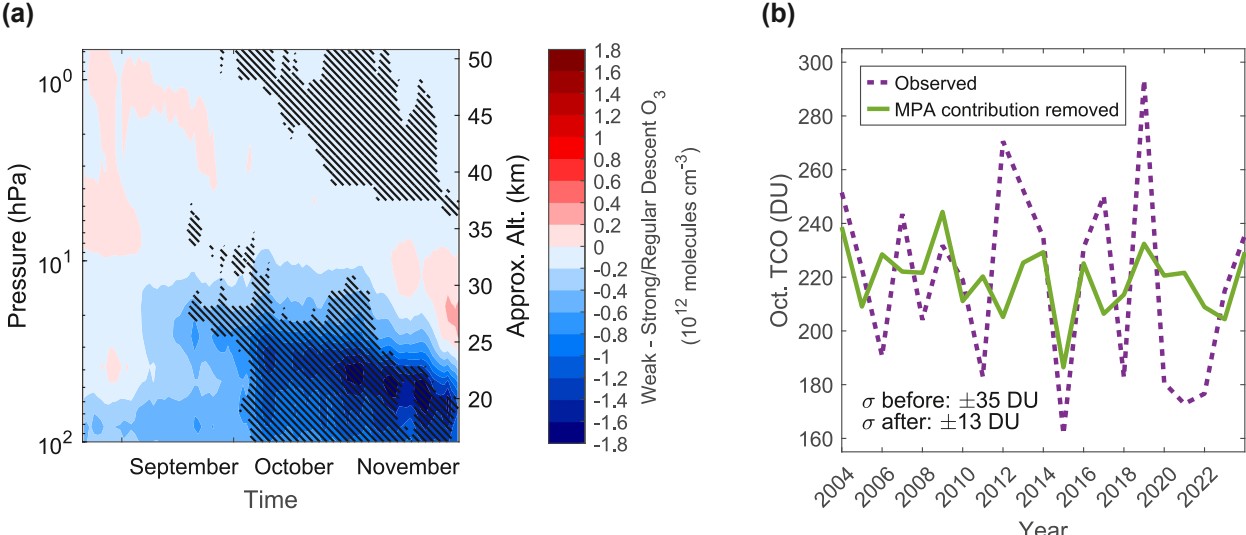

**Figure 9. Mesospheric Parcel Altitude (MPA) linked with October ozone hole outcomes. (a)** Composite difference of the daily averaged polar cap (65°S–82°S) MLS/Aura ozone ($O_3$) number density between the Weak and Strong/Regular mesospheric descent years (as listed in Table 2), from August 21st through November for 2004–2024. Hatching indicates points with significant negative correlation with the MPA ($|r| \geq 0.7$, significant at the 95% level). **(b)** October TCO from SBUV MOD before (purple, dashed lines) and after (green, solid lines) removing the contribution from a simple linear fit with the MPA. The standard deviation before and after accounting for the MPA are indicated with the $\sigma$ symbols on the figure.

mesospheric air parcel has spent prolonged time in the stratospheric polar vortex, suggesting that the interaction between the vertical descent pattern and horizontal effects within the inner-vortex (as reflected by the observable CO tracer) is key for explaining for ozone values.

Daily changes in springtime $O_3$ VMR profiles in Fig. 4 provide the first clue to the mechanism by which descending mesospheric air leaves its imprint on high-latitude ozone. During both Major (Fig. 4a) and Non-Major (Fig. 4b) ozone hole years, we find a positive-negative-positive pattern which follows the trajectory of the mesospheric parcel starting prior to September (the parcel location is shown by the black CO contour lines overlaid on both panels). The small negative pattern at the centre of the parcel is likely associated with the downwards movement of the $O_3$-poor, CO-rich envelope of air. The large-

magnitude, positive regions both above and below the parcel indicate that daily enhancements of $O_3$ ($\sim 0.02 \, \mathrm{ppmv \, day^{-1}}$) are occurring on the parcel boundaries. Similar between both panels, there is a prominent negative daily change in $O_3$ below 25 km from late-August through September where active $\mathrm{ClO_x}$-$\mathrm{BrO_x}$ depletion of $O_3$ is known to occur each year (Santee et al., 2008; WMO, 2022). Remarkably, by the time October begins and the mesospheric parcel is arriving at the lower stratosphere, nearly all altitudes below 27 km have transitioned to daily increases in $O_3$ which are instead characteristic of the behaviour at the

lower boundary of the parcel. The pattern of increasing $O_3$ at the upper and lower boundary of the descending air parcel must

arise from either $O_3$ production or transport of $O_3$-rich air to higher latitudes. From the evidence provided by the model, we postulate that $O_3$ production is neither significant nor varied enough below 27 km to explain the observed patterns in Fig. 4.

We use the daily CESM2 SD-WACCM chemical reaction rates presented Figs. 5-6 to confirm that the rate of net positive $O_3$ production below $\sim$27 km is negligible for the entire spring (see the relationship between the $O_x$ production and the multiple $O_x$ loss mechanisms which overlap to lower altitudes in Fig. 5). While there is extensive chemical activity above 27 km, $O_3$ transport here must be nearly comparable in magnitude to produce the daily delta patterns as shown in Fig. 4. Fig. 6a–b provide a picture of the corridor of horizontal $O_3$ transport that follows the trajectory of the descending mesospheric air parcel and dominates the resupply of ozone between 17–27 km. This transport-dominated region beneath the centre of the descending mesospheric parcel in October extends to approximately 70°S (see Fig. 6c–d) and shows a distinctly weaker signature during Major ozone hole years. Prior studies have found evidence of reduced mid-latitude ozone resupply during years with reduced planetary wave activity (with reduced wave activity being a main driver of a stabler, colder vortex) (Huck et al., 2005; McLandress et al., 2010; Orr et al., 2012; Klekociuk et al., 2021). Our results support the idea of large-scale horizontal transport into the interior of the polar vortex beginning in October, about a month sooner than expected during vortex breakdown in November and December (Bowman, 1990, 1993; Sutton, 1994).

We consider the utility of using the MPA metric to track the dynamical conditions of the inner-vortex. Kawamoto and Shiotani (2000) find that the polar vortex shifts to lower altitudes in years with strong planetary wave activity, resulting in faster descent within the vortex. We produce complementary findings, with Fig. 7a showing strong negative correlations between the ERA5 accumulated SH EHF (from March–October) and the MERRA-2 zonal wind strength and altitude (from September–October). The ERA5 EHF also correlates with the MLS/Aura MPA ($r = -0.83$), indicating that the descending mesospheric air parcel remains at higher altitudes when planetary wave driving is weaker.

When looking at the MERRA-2 zonal wind metrics directly, the magnitude of the correlation with the MPA grows ($r \geq 0.9$ for both). This would suggest that the MPA remains higher during years with faster, higher-altitude zonal winds. Based on these links, the trend pointed out in Kessenich et al. (2023) of reduced mesospheric descent between 2004–2022 is likely the product of a stronger polar vortex in the latter half of the time series, potentially preconditioned by reduced planetary wave driving. This dynamical explanation is in agreement with recent findings of a delayed SH vortex breakdown in 2020 (Lecouffe et al., 2022), with year-to-year variation in vortex evolution driven by the solar cycle, the Quasibiennial Oscillation, and the El Niño–Southern Oscillation (Baldwin and Dunkerton, 1998; Li et al., 2016; Domeisen et al., 2019; Lecouffe et al., 2022).

To build a proxy between the MPA (and effectively the dynamical conditions of the inner-vortex) and ozone transport, we test for correlations between modelled MPA, observed MPA, and model-estimated ozone transport in Fig. 7. We confirm that the MPA shows a robust relationship with ozone transport, and modelled and observed MPA show a robust linear relationship with each other. Notably, of all metrics tested in Fig. 7a, the MPA produces the strongest correlation with ozone transport and offers perhaps the most directly-observable proxy. We apply the linear relationship between MLS/Aura MPA and model-estimated ozone transport identified in Fig. 7b to produce Equation (1) which estimates the magnitude of horizontal ozone transport poleward of 70°S and between 17–27 km in October. The results from our MPA calculations and predicted transport magnitudes are presented in Table 2, and from this we can conclude that transport is reduced when mesospheric descent is

Weak and enhanced when mesospheric descent is Strong. The dynamical evolution of the ozone hole is linked with the MPA, with 63% of October TCO variation removed when the MPA is accounted for in Fig. 9b. Not only does the MPA account for the low TCO values during Major ozone hole years, but it also captures the high TCO values during years such as 2012 and 2019. For the years 2020–2022 in particular, we estimate that the TCO would be on average $\sim$40 DU higher without the reduction in transport that likely occurred due to vortex dynamics (as proxied by the MPA). Alternatively, the TCO values for 2023 and 2024, the two most recent Regular descent years, are relatively unchanged after accounting for the MPA. We note that observations of mesospheric descent into early November are needed to calculate the MPA for a given year, meaning that this metric functions primarily as a diagnostic (rather than a seasonal predictive) tool.

## 5 Conclusions

An accurate attribution of chemical versus dynamical contributors to Antarctic ozone variability is especially important when interpreting recent ozone holes (2020–2022) which have remained large and deep late into the spring season. Early October is particularly interesting, as it represents a transitional period when the chemical evolution of the ozone hole gives way to predominately dynamical drivers. To improve our diagnostic capability for October TCO, we use MLS/Aura CO observations to track the springtime descent of a large parcel of mesospheric air that descends through the polar vortex as the ozone hole develops in the lower stratosphere below. We define a new metric based on the location of this mesospheric parcel at the end of October, termed the Mesospheric Parcel Altitude (MPA), which we link to the dynamical state of the polar vortex and large-scale poleward transport of ozone. From this metric, we classify ozone hole years into characteristic descent types and provide a formula (Eq. 1) that can be used to estimate the magnitude of October ozone transport poleward of 70°S and between 17–27 km (see Table 2). An ozone hole with Weak mesospheric descent (MPA >26.9 km) is likely to have reduced ozone transport and remain Major through October, and one with Strong mesospheric descent (MPA <24.6 km) is likely to have enhanced transport and be Minor. When applied as a proxy for polar cap TCO, the majority of year-to-year variation during October can be explained by the MPA.

The circulation effects of large and long-lived Antarctic ozone holes are widespread, propagating across the entire SH and modulating tropospheric climate (Son et al., 2010; Thompson et al., 2011; Polvani et al., 2011; Orr et al., 2012), with middle stratospheric ozone loss dominating forcing signals (Keeley et al., 2007). Secondarily, UV levels on the surface of Antarctica are worsened when the ozone hole remains deep late into the spring (Barnes et al., 2022; Cordero et al., 2022), exacerbating an already fragile ecosystem with declining ice sheet volume (Purich and Doddridge, 2023). While long-term trends in stratospheric chlorine loading are relatively predictable, long-term circulation trends due to climate change are less so. With the caveat that satellite monitoring systems continue to observe the CO tracer in the future, the MPA provided in this work serves as sensitive diagnostic measure of how the interior of the polar vortex responds to large-scale circulation forcing.

# Appendix A

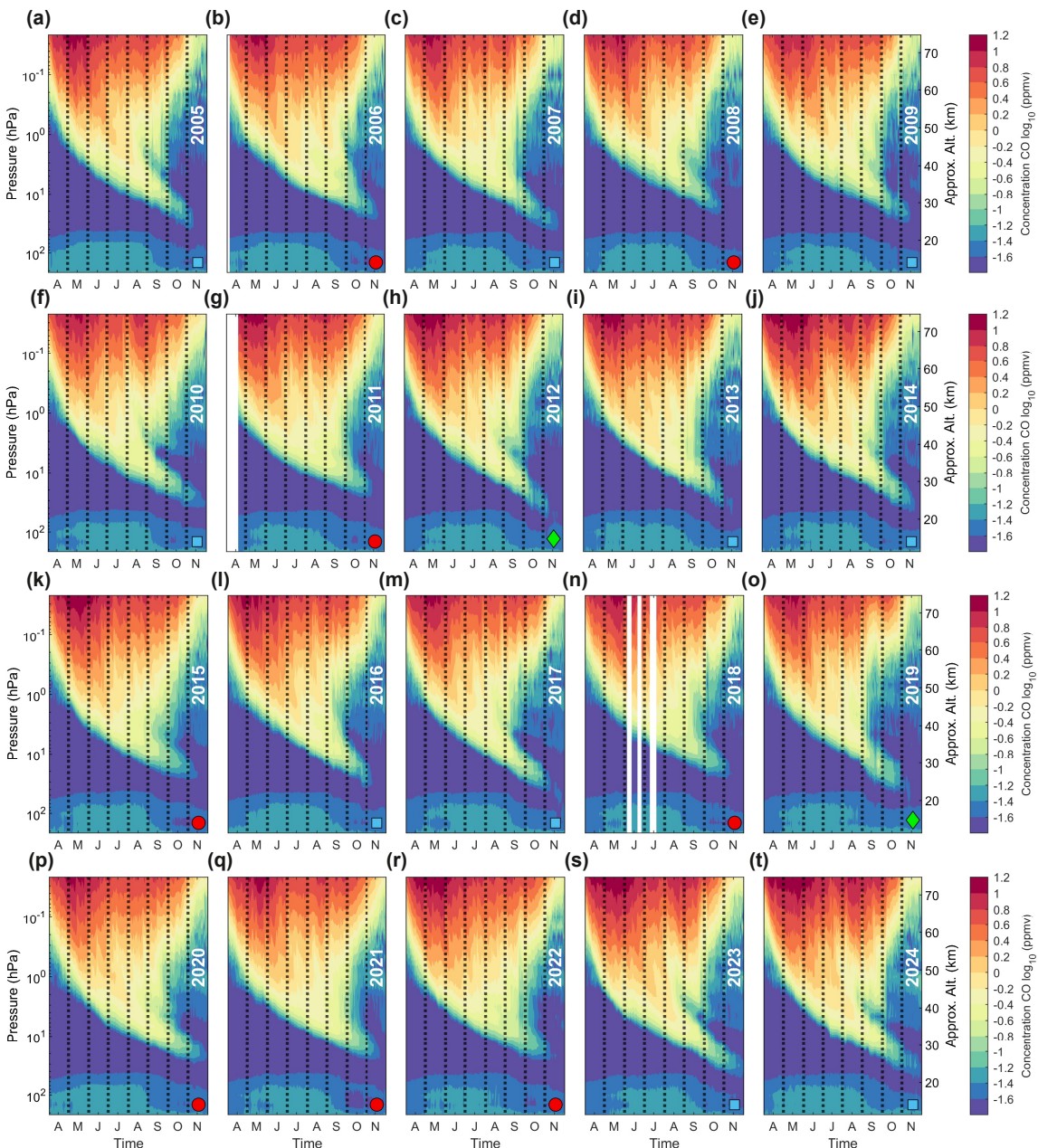

**Figure A1. MLS/Aura carbon monoxide (CO) observations.** Daily observations of zonally averaged high-latitude (75°S–82°S) CO VMR profiles (shown with a logarithmic scale) from April to November. Years, as labelled in each panel, are presented consecutively from **(a)** 2005 to **(t)** 2024. Minor ozone hole years are labelled with green diamonds, Moderate ozone hole years are labelled with cyan squares, and Major ozone hole years are labelled with red circles (see Table 1). Months on the $x$-axis are indicated with their corresponding first letter and are separated by black dashed lines.

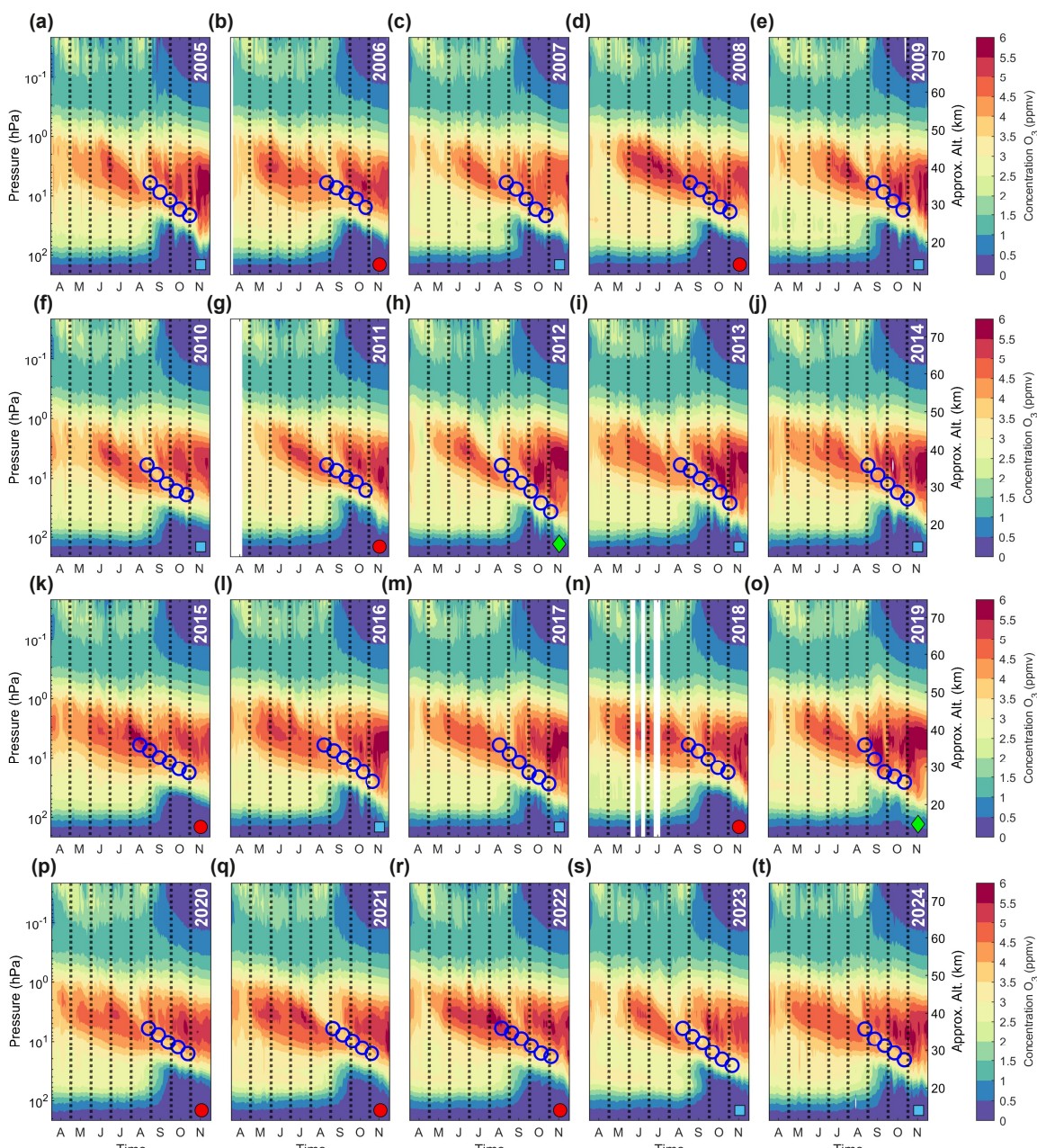

**Figure A2. MLS/Aura ozone (O$_3$) observations.** Daily observations of high-latitude-core O$_3$ VMR profiles from April to November. Years, as labelled in each panel, are presented consecutively from **(a)** 2005 to **(t)** 2024. Minor ozone hole years are labelled with green diamonds, Moderate ozone hole years are labelled with cyan squares, and Major ozone hole years are labelled with red circles (see Table 1). Blue circles are overlaid at the altitude of the corresponding day's carbon monoxide profile maximum. Months on the $x$-axis are indicated with their corresponding first letter and are separated by black dashed lines.

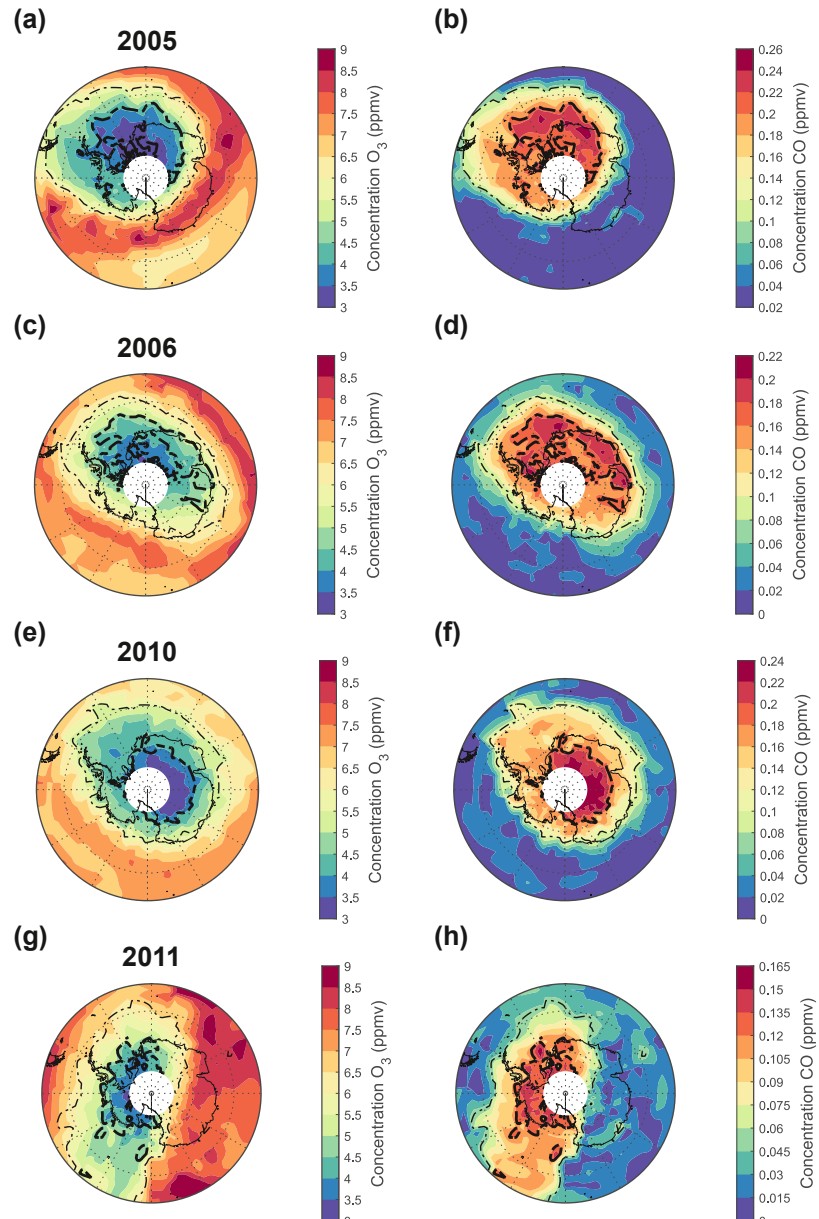

**Figure A3. Maps of MLS/Aura ozone** ($O_3$) **and carbon monoxide (CO) observations.** Side-by-side observations of **(a)** $O_3$ and **(b)** CO at 10 hPa (∼31 km) are shown on the 27th September 2005. The day is selected to best show the mesospheric air parcel (tracked with MLS/Aura CO, see Methods) at 10 hPa. The $O_3$ and CO colour contours (shown in VMR) are scaled for best contrast. Corresponding maps are shown for **(c)–(d)** the 7th October 2006, **(e)–(f)** the 25th September 2010, and **(g)–(h)** the 12th October 2011. To aid in spatial comparison between each pair of maps, a thin black dash-dot line is drawn around the 50th percentile CO contour, and a thick dashed line is drawn around the 85th percentile CO contour. Latitude grid lines are shown for 60°S and 75°S, and no data is plotted beyond 82°S.

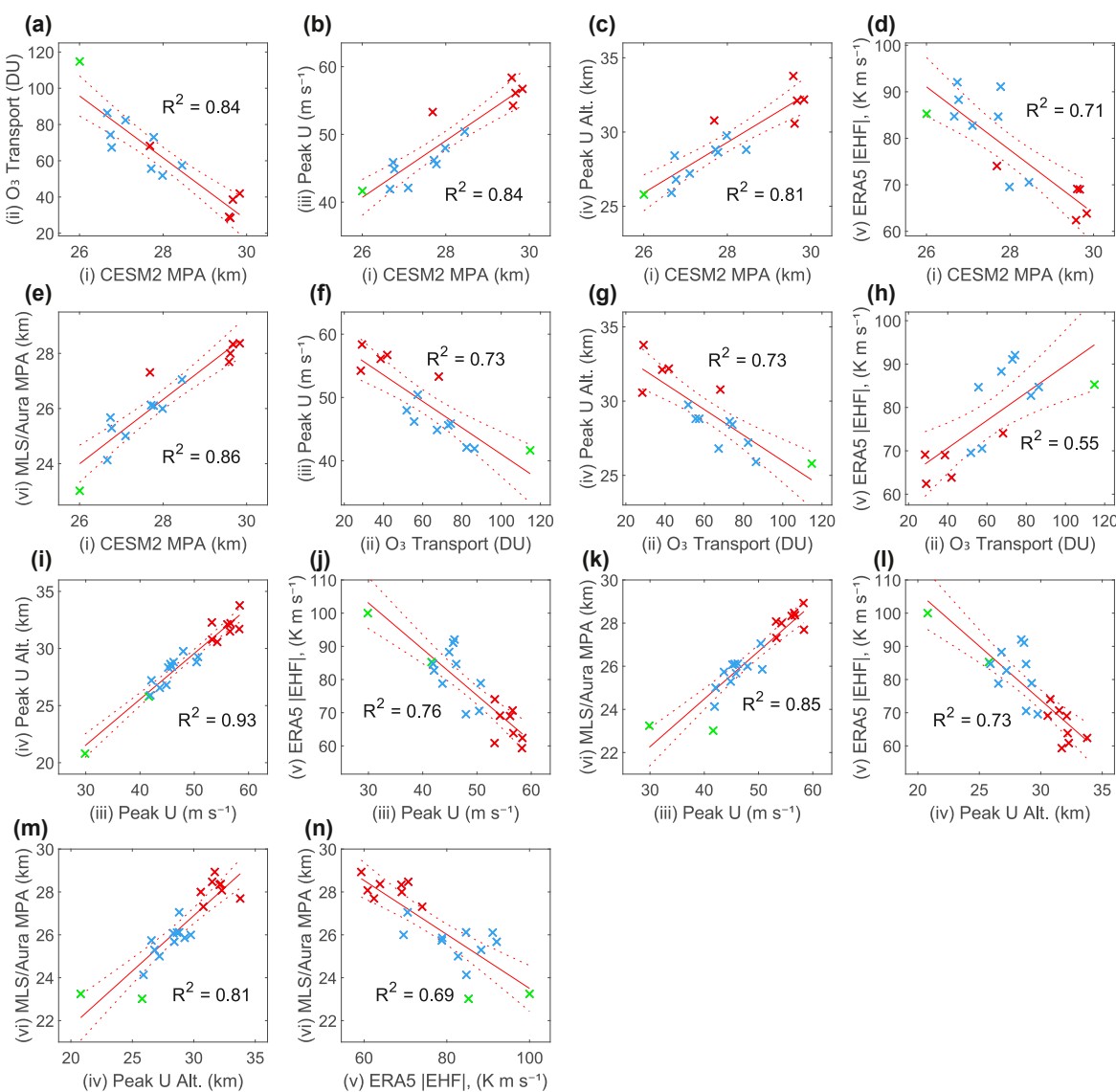

**Figure A4. The relationship between dynamical proxies and ozone ($O_3$) transport.** Linear fits are performed between (i) CESM2 MPA (km) for 2005–2018, (ii) CESM2 horizontal $O_3$ transport between 70°S–90°S from 17–27 km (DU) for 2005–2018, (iii) the average Sept.–Oct. strength of the peak 45°S–75°S MERRA-2 zonal wind (U) ($m\,s^{-1}$) for 2004–2024, (iv) the average Sept.–Oct. altitude of the peak 45°S–75°S MERRA-2 zonal wind (U) (km) for 2004–2024, (v) the magnitude of the Mar.–Oct. accumulated ERA5 eddy heat flux (|EHF|) ($K\,m\,s^{-1}$) for 2004–2023, and (vi) observed MLS/Aura MPA (km) for 2004–2024. Red lines show the result from each fit and its 95% confidence bounds. The corresponding $R^2$ value is given. Years are colour-coded according to the categories in Table 1: Green = Minor, Blue = Moderate, and Red = Major ozone holes.

*Code availability.* CESM2 source code is distributed through a public subversion code repository (https://www.cesm.ucar.edu/models/cesm2).

*Data availability.* All data used in this study is freely available from the following sources:

The MLS/Aura Level 2 Ozone Mixing Ratio Version 5 data are available at https://disc.gsfc.nasa.gov/datasets/ML2O3_005/summary?keywords=mls%20o3 (Schwartz et al., 2020).

    The MLS/Aura Level 2 CO Mixing Ratio Version 5 data are available at https://disc.gsfc.nasa.gov/datasets/ML2CO_005/summary?keywords=mls%20co (Schwartz, M., Pumphrey, H., Livesey, N., and Read, W., 2020).

    The SBUV Merged Ozone Data set (MOD) version 8.7 is available at http://acdb-ext.gsfc.nasa.gov/Data_services/merged/ (Frith et al., 475 2014).

    The MERRA-2 3-dimensional 6-hourly data collection is available at https://disc.gsfc.nasa.gov/datasets/M2I6NPANA_5.12.4/summary?keywords=merra2 (Global Modeling and Assimilation Office (GMAO), 2015).

    The ERA5 accumulated EHF data is available at https://www.iup.uni-bremen.de/OREGANO/index_proxy.html.

*Author contributions.* HEK and AS planned the study. HEK ran the CESM2 simulations. HEK analysed the data and prepared the figures 480 with suggestions from AS, DS, CJR, and MW. HEK wrote the manuscript with comments from AS, DS, CJR, and MW.

*Competing interests.* The authors declare no competing interests

*Acknowledgements.* The CESM project is supported primarily by the U.S. National Science Foundation.

    The authors wish to acknowledge the use of New Zealand eScience Infrastructure (NeSI) high performance computing facilities and consulting support as part of this research. New Zealand's national facilities are provided by NeSI and funded jointly by NeSI's collaborator 485 institutions and through the Ministry of Business, Innovation & Employment's Research Infrastructure programme. URL: https://www.nesi.org.nz.

    This research was undertaken with the assistance of resources from the National Computational Infrastructure (NCI Australia), an NCRIS enabled capability supported by the Australian Government.

    HEK and AS were partially funded by the New Zealand Ministry of Business, Innovation & Employment Endeavour fund Smart Ideas 490 project PROP-76111-ENDSI-UOO (Contract UOOX2106).

    The authors would like to thank Stacey Frith for providing the preliminary 2024 SBUV MOD results for use in this publication.

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
