# Peer review of "Quantifying the contribution of transport to Antarctic springtime ozone column variability"

_EGUsphere, 2025_

## Referee Comment (RC2)

**Review of "Quantifying the contribution of transport to Antarctic springtime ozone column variability" by H.E. Kessenich et al. (2025).**

This paper seeks to identify drivers of interannual variability of Antarctic ozone in October. Elaborating on the results of Kessenich et al. (2023), the authors link the variability of polar cap TCO with the intensity of mesospheric descent. The paper introduces a novel diagnostic of the latter, the mesospheric parcel altitude (MPA) based on tracking the daily evolution of the polar cap CO. It is shown that MPA is highly correlated with the TCO in October.

The paper is well-written with clear figures and comprehensive captions. MPA is certainly an interesting new metric of mesospheric/stratospheric descent. While the use of CO as a transport tracer of mesospheric air into the polar vortex has a long history, it is good to look at things from a different perspective. I find the discussion of the descending layered pattern of high-low-high ozone especially interesting, although a bit confusing at times, at least to me (see my general comment concerning horizontal vs. vertical transport). However, I do have several serious concerns related to the main message of the paper and some of the methods used. These are delineated in my general comments below. My main criticism is that this paper appears to single out year-to-year fluctuations in mesospheric transport as the primary driver of interannual variability of October TCO but never really demonstrates that this is the case. The analysis is based on correlations between metrics like MPA and TCO. But don't these correlations arise from the fact that both diagnostics (and several others) are correlated with wave activity, which is a primary driver of TCO (as is well established), and not from a causal relationship between mesospheric descent and TCO? It's entirely possible that I grossly misunderstood this point of the paper. I will be happy to be corrected.

Much of it may simply be a matter of rephrasing parts of the manuscript, but since the problem is with what appears to be the main message of this work, I think it's fair to say that I'm asking for major revisions.

**General comments**

1. The introduction states: "*We seek to formalize the link identified in Kessenich et al. (2023), ultimately hypothesising that October TCO outcomes are dominated by descent-modulated horizontal transport effects.*" The way I read it, the claim is that variability in ozone transport in the middle-to-upper portion of the vortex (mesospheric descent reaches down to 25 km / 20 hPa or so) drives interannual variability of TCO in October. I don't think this is actually demonstrated in the paper and I don't believe it is correct. I've tried to estimate how much of the overall year-to-year TCO variability can be attributed to variability of ozone in the part of the vortex affected by

mesospheric descent. I did a quick calculation based on M2-SCREAM, which is basically MLS ozone data. The standard deviation of the 2004–2024 October mean polar cap partial ozone column between 26 hPa and the top of the atmosphere is only 5.43 DU, with the maximum range of ~20 DU. In contrast, for the polar cap TCO, these numbers are as high as 24 DU (the paper finds it to be 35 DU, line 352 of the manuscript) and 112 DU, respectively. Furthermore, the correlation between October TCO and the 26 hPa–TOA partial column is only about 0.55. If this is correct then only a fraction of TCO variability can be "blamed on" what's happening in the mid-to-upper polar stratosphere. This makes sense because of the relatively low air density at those altitudes compared to the lower stratosphere. So why is MPA such a good predictor of TCO? It is known that the main driver of variability of ozone hole sizes and ozone mass deficit is wave activity and polar temperature (also linked to wave activity) (Newman et al., 2004; Huck et al., 2005, already cited, among many other publications including several Ozone Assessments back to the 2006 one). I understand that many of those studies did not focus specifically on October but the point is that wave activity affects the vortex temperature, diabatic descent within the vortex (and ozone resupply), the vortex size and stability, all of which impact the amount of springtime ozone over the high latitudes throughout the stratosphere. Your results show that MPA is highly correlated with wave activity. This certainly makes MPA a good predictor of October TCO but does not establish a causal relationship. I would like to see some clarification here.

2. **Averaging**. First, the procedure for calculating the inner-vortex ozone profile seems very complicated. Why is it done this way and why does it work? How do you define inner vortex? Why not use a dynamical definition of the polar vortex instead, e.g. based on reanalysis PV (Manney et al., 1994; Lawrence et al., 2018)? Other formulations also exist (Nash et al. 1996). Note that PV-based approaches can be refined to distinguish between the vortex core and the outer vortex. It seems circular to use an ozone-based definition of the vortex to estimate vortex ozone. Second, I'm not sure if I understand why CO is averaged zonally instead of within the vortex. Section 2.1 states: "*as stratospheric CO concentrations are very low beyond the edge of the polar vortex, we use a simple high-latitude (75°S–82°S) zonal average of daily CO observations.*" But the fact that CO concentrations are near zero outside the vortex is all the more reason to average CO within the vortex if we want to use it as a diagnostic of air descent. When averaged zonally, any dilution from extra-vortex air will result in a large decrease in CO that has nothing to do with the descent but, instead, is a consequence of sharp gradients across the vortex edge. It is also inconsistent with the way "inner vortex" ozone is averaged. Again, it's possible that I misunderstood something.

3. **Transport**. When seeking to explain the three-layered pattern of the "ozone corridor" the authors disentangle chemistry from transport, and conclude, based on model simulations, that the pattern and its interannual variability arise from **horizontal** transport and its variability. But the "horizontal" part is never explained. Polar ozone has significant vertical gradients, which are acted upon by diabatic descent. Why is it assumed that only horizontal transport variability is important?

**Specific comments**

L114. To quantify the "severity of ozone depletion" one would need to estimate chemical loss throughout the stratosphere. This is not what is done here. Or is the term "depletion" used in a different sense?

LL113-116. Polar cap TCO is indeed one of the standard metrics of Antarctic ozone. Other metrics are the ozone hole area and ozone mass deficit (OMD), and these diagnostics are more directly related to ozone holes as opposed to overall high-latitude ozone. Is TCO sufficiently highly correlated with area and OMD to be used as the sole diagnostic here (see, e.g., See Wang et al. (2025) especially their discussion around Extended Data figures 8 and 8)? Would the classification in Table 1 look the same if the other metrics were used?

L128. There are several ways of doing specified dynamics (various flavors of nudging, replay, etc.). How exactly is it done in this case? Is there a reference for it?

L130. Why not all the way to 2024?

L149. No need to define MERRA-2 again (already defined in LL123-124).

L177. Consistent with what?

L182-183. Why? Number density is not a conserved quantity.

Figure 4. The red circle in panel (a) is barely visible against the red shading.

L231. Why not vertical transport? October ozone maximum over Antarctica is at about 6–7 hPa. At least the lower maximum in Fig. 4 could arise from diabatic descent.

L250. I'm not sure if I see that. To me loss due to NOx reaches down to the lower of the two maxima in $O_3$ tendency while production stops closer to the upper one.

L278.How is the "inner boundary of the stratospheric polar vortex" defined? Also, again why does it have to be horizontal transport? I'm not saying that it isn't, but it would be good to see some explanation.

L284. This is the only place where the term "outer boundary" is used. Similarly to "inner boundary" and "inner vortex", it is never defined. Please, provide those definitions in the methods section.

LL343-349. This is related to my general comment #1. The way I read it Figure 8a, it shows that most of the TCO difference between Weak and Strong/Regular descent years comes from the lower stratosphere. This makes sense because years when the vortex is relatively undisturbed due to weak wave activity tend to have stronger / more widespread heterogeneous depletion in June–September and, consequently less ozone in the lower stratosphere in October/November. In contrast, middle and upper stratospheric ozone difference, while of the same negative sign, contributes very little to the overall difference. The efficacy of MPA as a diagnostic here results from the fact that MPA is highly correlated with wave activity, not from a direct contribution of upper stratospheric transport to the size of the ozone hole. Is that a correct interpretation? At the risk of repeating myself, I think this really needs more discussion because as the paper is written one gets the impression that mesospheric transport controls the TCO (as opposed to be simply correlated with it) in October, which I don't think is correct.

LL350-354. Strictly speaking a metric can't "contribute" to a physical process. I interpret this statement as saying that variability of mesospheric descent (which MPA is a measure of) is a major contribution to the size of the ozone holes. But that's not the case. All this calculation shows is that MPA is a good diagnostic of the variability of October TCO, which is a fine result. But it does not demonstrate that mesospheric descent is responsible for this variability. Conversely, one could also say that TCO variability is a good metric of upper-atmosphere transport. It's a correlation vs. causality type of thing. I really suggest that this be clearly stated.

LL367-368. Actually, since CO is averaged zonally, not within the vortex, this doesn't tell us how much time the parcel has spent in the vortex. I would appreciate a comment on this.

Thank you,
Kris Wargan

**References**

Lawrence, Z. D., Manney, G. L., and Wargan, K.: Reanalysis intercomparisons of stratospheric polar processing diagnostics, Atmos. Chem. Phys., 18, 13547–13579, https://doi.org/10.5194/acp-18-13547-2018, 2018.

Manney, G. L., Zurek, R. W., O'Neill, A., & Swinbank, R. (1994). On the motion of air through the stratospheric polar vortex. Journal of the Atmospheric Sciences, 51(20), 2973–2994. https://doi.org/10.1175/1520-0469(1994)051<2973:OTMOAT>2.0.CO;2

Nash, E. R., P. A. Newman, J. E. Rosenfield, and M. R. Schoeberl (1996), An objective determination of the polar vortex using Ertel's potential vorticity, J. Geophys. Res., 101(D5), 9471–9478, doi:10.1029/96JD00066.

Newman, P. A., S. R. Kawa, and E. R. Nash (2004), On the size of the Antarctic ozone hole, Geophys. Res. Lett., 31, L21104, doi:10.1029/2004GL020596.

Wang, P., Solomon, S., Santer, B.D. et al. Fingerprinting the recovery of Antarctic ozone. Nature 639, 646–651 (2025). https://doi.org/10.1038/s41586-025-08640-9

---

## Author Response (AR1)

**Author response to reviewer comments**

We would like to thank both reviewers for their comments on the manuscript. Our detailed responses are included below.  In addition to changes reflecting specific comments, some minor changes were made to improve overall clarity. These are listed at the end of the document.  We also updated the TCO and MPA thresholds so that they are more straightforward mathematically. This resulted in very minor updates of some values in Tables 1 and 2, as listed at the end of our responses.

**Reviewer #1:**

**Overall reviewer comment:**

I have only one suggestion: If possible, it would be helpful to have a conceptual picture / figure that shows how the mesospheric descent and the transport into the polar region are connected. Fig. 6 goes a little bit into that direction, but a conceptual picture early on, would probably help for easier understanding of the paper, which then goes into quite a lot of technical detail. Apart from that: a great and well-written paper. Congratulations!

**Author response:**

We have added a conceptual diagram to the end of Section 3.5, now Figure 8. At this point in the paper, all important terminology has been introduced and links to vortex dynamics have been presented. The diagram pulls the full picture of mesospheric descent and horizontal transport together. We also will be adding an animated version of this figure as a Supplement.

**Reviewer #2:**

**Overall reviewer comment:**

My main criticism is that this paper appears to single out year-to-year fluctuations in mesospheric transport as the primary driver of interannual variability of October TCO but never really demonstrates that this is the case. The analysis is based on correlations between metrics like MPA and TCO. But don't these correlations arise from the fact that both diagnostics (and several others) are correlated with wave activity, which is a primary driver of TCO (as is well established), and not from a causal relationship between mesospheric descent and TCO? It's entirely possible that I grossly misunderstood this point of the paper. I will be happy to be corrected. Much of it may simply be a matter of rephrasing parts of the manuscript, but since the problem is with what appears to be the main message of this work, I think it's fair to say that I'm asking for major revisions.

*Author response:*

In this manuscript, we demonstrate that transport into the middle/lower polar stratosphere during October is correlated with the MPA (Fig. 7), and this transport (proxied by the MPA) dominates ozone variability during October (Fig. 9b). Model calculations are used to separate transport from chemistry (Fig. 6), and the MPA reflects the dynamical conditions within the polar vortex (Fig. 7a). The language used in the Abstract, Discussion, and Conclusions reflects these findings. For more specific details, please see the responses the Comments #1, #4, #17, and #18.

**General comments:**

1. The introduction states: "We seek to formalize the link identified in Kessenich et al. (2023), ultimately hypothesising that October TCO outcomes are dominated by descent-modulated horizontal transport effects." The way I read it, the claim is that variability in ozone transport in the middle-to-upper portion of the vortex (mesospheric descent reaches down to 25 km / 20 hPa or so) drives interannual variability of TCO in October. I don't think this is actually demonstrated in the paper and I don't believe it is correct. I've tried to estimate how much of the overall year-to-year TCO variability can be attributed to variability of ozone in the part of the vortex affected by mesospheric descent. I did a quick calculation based on M2-SCREAM, which is basically MLS ozone data. The standard deviation of the 2004–2024 October mean polar cap partial ozone column between 26 hPa and the top of the atmosphere is only 5.43 DU, with the maximum range of ~20 DU. In contrast, for the polar cap TCO, these numbers are as high as 24 DU (the paper finds it to be 35 DU, line 352 of the manuscript) and 112 DU, respectively. Furthermore, the correlation between October TCO and the 26 hPa–TOA partial column is only about 0.55. If this is correct then only a fraction of TCO variability can be "blamed on" what's happening in the mid-to-upper polar stratosphere. This makes sense because of the relatively low air density at those altitudes compared to the lower stratosphere. So why is MPA such a good predictor of TCO? It is known that the main driver of variability of ozone hole sizes and ozone mass deficit is wave activity and polar temperature (also linked to wave activity) (Newman et al., 2004; Huck et al., 2005, already cited, among many other publications including several Ozone Assessments back to the 2006 one). I understand that many of those studies did not focus specifically on October but the point is that wave activity affects the vortex temperature, diabatic descent within the vortex (and ozone resupply), the vortex size and stability, all of which impact the amount of springtime ozone over the high latitudes throughout the stratosphere. Your results show that MPA is highly correlated with wave activity. This certainly makes MPA a good predictor of October TCO but does not establish a causal relationship. I would like to see some clarification here.

*Author response:*

The MPA is indeed intended to function as a diagnostic of transport effects. We do not wish to give the impression that the ozone transport is from the mesosphere, rather the MPA captures the overall polar response to forcing (e.g. waves). We fully agree that the wording in the introduction should be reflective of the outcomes. We have revised this sentence so that it now reads,

*"concluding that descent from the mesosphere serves as a diagnostic indicator of vortex dynamics and horizontal ozone transport."*

Another example where the text has been clarified is at L. 306-308 in the final manuscript,

*"We therefore hypothesise that the MPA fundamentally reflects the strength of the accumulated EHF and the resulting inner-vortex dynamical conditions, which then control the magnitude of horizontal $O_3$ transport towards the pole."*

In addition to the specific two issues addressed above, we have reviewed the language used in the manuscript to ensure the text does not imply causation.

On the comment relating to pressure levels, the transport estimates in Table 2 indicate that the region of the atmosphere where transport is being calculated is between 17 and 27 km (~20-90 hPa). This altitude range is stated several times in the text (L. 11, 288, 293, 298, 310, 336, 350, and 410), which we hope is clear enough to distinguish from the "mid-to-upper" stratospheric range. We fully agree that transport of ozone from above ~20 hPa is not sufficient to cause large variations in TCO values.

2. Averaging. First, the procedure for calculating the inner-vortex ozone profile seems very complicated. Why is it done this way and why does it work? How do you define inner vortex? Why not use a dynamical definition of the polar vortex instead, e.g. based on reanalysis PV (Manney et al., 1994; Lawrence et al., 2018)? Other formulations also exist (Nash et al. 1996). Note that PV-based approaches can be refined to distinguish between the vortex core and the outer vortex. It seems circular to use an ozone-based definition of the vortex to estimate vortex ozone.

*Author response:*

We now recognise that the term "inner-vortex" is not the best term for this ozone profile time series, as it has an established meaning. To address this, we have changed the wording throughout so that the goals of this method are not confused with other similar analyses which seek to only include air residing in the inner polar vortex. We now use "high-latitude-core" selection, updated throughout. The description in Section 2.1 has been updated to better reflect the goals of the method:

- L. 64-65: *"select an altitude-dependent approximation of the highest latitude air within the core of the ozone hole"*
- L. 71-73: *"The outcome of these steps is a dataset which holds approximately the same surface area at every vertical level, with data points corresponding to the highest latitudes within the ozone hole. This dataset will be used for visualization purposes only."*

The choice to use this method, rather than a PV-based approach, was motivated by the following: (1) Alignment with the CO profile format: For the purposes of calculating the MPA, the units of hPa/km were best suited for the vertical axis of the CO profiles. We wanted the vertical units of the $O_3$ dataset to match. (2) Retaining data in the middle/upper stratosphere: We tested PV-based inner-vortex selections (such as that provided by the MLS/Aura L3 datasets) and found that a large portion of the middle/upper stratosphere is excluded from these datasets. With the high-latitude-core method, we retain as much data as possible while still isolating ozone within the core of the ozone hole. We also note that these $O_3$ profiles are used only for visualization purposes and are not used in any outputs from the manuscript (such as the MPA, transport estimates, and ozone hole classifications).

3. I'm not sure if I understand why CO is averaged zonally instead of within the vortex. Section 2.1 states: "as stratospheric CO concentrations are very low beyond the edge of the polar vortex, we use a simple high-latitude (75°S–82°S) zonal average of daily CO observations." But the fact that CO concentrations are near zero outside the vortex is all the more reason to average CO within the vortex if we want to use it as a diagnostic of air descent. When averaged zonally, any dilution from extra-vortex air will result in a large decrease in CO that has nothing to do with the descent but, instead, is a consequence of sharp gradients across the vortex edge. It is also inconsistent with the way "inner vortex" ozone is averaged. Again, it's possible that I misunderstood something.

**Author response:**

Our priority with the creation of the MPA was to keep it as easy as possible to replicate. We found that the peak-finding methodology of the daily descent-tracking CO time series was robust to dilution effects from air outside the vortex (as described in L. 87-101). Furthermore, we found that the 75°S–82°S selection captured a higher portion of the descending parcel in the upper stratosphere, not possible with traditional inner-vortex selections (such as that provided by MLS/Aura L3 datasets) which exclude a significant portion of air above the lower stratosphere. A sentence has been added to the text to provide further justification for this choice:

- L. 96-97: *"To keep this method easily replicable, we start with the simple high-latitude CO dataset (averaged from 75°S–82°S) rather than one based on the core of the ozone hole"*

4. Transport. When seeking to explain the three-layered pattern of the "ozone corridor" the authors disentangle chemistry from transport, and conclude, based on model simulations, that the pattern and its interannual variability arise from horizontal transport and its variability. But the "horizontal" part is never explained. Polar ozone has significant vertical gradients, which are acted upon by diabatic descent. Why is it assumed that only horizontal transport variability is important?

*Author response:*

The text discussing Figure 6 starting at L. 282 has been revised to provide a more thorough interpretation of the transport patterns shown in the figure. To specifically discuss the horizontal nature of the transport, the following statements have been added,

*"For every modelled year, we find that (1) $O_x$ loss is greater than $O_x$ production and (2) transport is positive, at all altitudes below 43 km and poleward of 70°S. From this, we can exclude the likelihood of ozone resupply from higher altitudes and conclude that the pattern between 17–27 km is most likely the result of horizontal ozone transport."*

Additionally, we have created an animation to include in as a supplementary material. The animation shows the ozone layer in units of number density through the spring (along with the zonal mean zonal wind and CO in units of VMR). The horizontal nature of the ozone transport is easier to see in the animation. The quantity of ozone molecules above ~20 hPa is low enough that vertical transport cannot significantly raise the ozone number density below this altitude.

**Specific comments:**

5. L114. To quantify the "severity of ozone depletion" one would need to estimate chemical loss throughout the stratosphere. This is not what is done here. Or is the term "depletion" used in a different sense?

*Author response:*

We have changed this from "*severity of ozone deletion*" → "TCO"

6. LL113-116. Polar cap TCO is indeed one of the standard metrics of Antarctic ozone. Other metrics are the ozone hole area and ozone mass deficit (OMD), and these diagnostics are more directly related to ozone holes as opposed to overall high-latitude ozone. Is TCO sufficiently highly correlated with area and OMD to be used as the sole diagnostic here (see, e.g., See Wang et al. (2025) especially their discussion around Extended Data

figures 8 and 8)? Would the classification in Table 1 look the same if the other metrics were used?

*Author response:*

We find that ozone hole area has a correlation of -0.97 with October TCO, and OMD has a correlation of -0.96 with October TCO (both metrics sourced from NASA Ozone Watch for 2004-2024). When area is used for the classification, categories in Table 1 are unchanged besides the year 2005, which moves from Moderate to Major. When OMD is used for the classification, the Major and Non-Major categories are unchanged, but there is some shift between Minor and Moderate. We choose October TCO due to its ability to capture the combined effect of chemistry within the vortex and dynamical ozone resupply.

7. L128. There are several ways of doing specified dynamics (various flavors of nudging, replay, etc.). How exactly is it done in this case? Is there a reference for it?

*Author response:*

We have added the following reference to L. 132 which provides more details on running WACCM with specified dynamics (SD-WACCM): Gettelman et al., 2019

8. L130. Why not all the way to 2024?

*Author response:*

Some of the forcing files needed for the model simulations were not consistently available for the full time period, thus we end the simulation on year 2018.

9. L149. No need to define MERRA-2 again (already defined in LL123-124).

*Author response:*

This definition has been removed.

10. L177. Consistent with what?

*Author response:*

This portion of the sentence has been removed, as it was redundant from the section above.

11. L182-183. Why? Number density is not a conserved quantity.

*Author response:*

We tested different units for quantifying the peak in CO concentration. While not conserved, the peak CO measured in number density undergoes a linear change in concentration as the mesospheric parcel descends during the month of October, unlike the peak measured in VMR. The relationship between peak CO and October TCO in Fig.

2a was found to be similar (low $R^2$) in all tests using alternative units of CO. We altered the text which describes this calculation in (a) at L. 185 for better clarity:

"*The average daily maximum in CO concentration during October*"

12. Figure 4. The red circle in panel (a) is barely visible against the red shading.

*Author response:*

We moved the circle location to make it more visible.

13. L231. Why not vertical transport? October ozone maximum over Antarctica is at about 6–7 hPa. At least the lower maximum in Fig. 4 could arise from diabatic descent.

*Author response:*

Given that our conclusions regarding the horizontal nature of the ozone transport in this region are not drawn until Figure 8, we have generalized this statement,

"*this pattern could be the result of either (1) chemical production of O₃, or (2) transport of O₃*"

14. L250. I'm not sure if I see that. To me loss due to $NO_x$ reaches down to the lower of the two maxima in $O_3$ tendency while production stops closer to the upper one.

*Author response:*

This statement is discussing the production (green line) versus the $NO_x$ loss (yellow line). The green line is overlapped by the yellow line, which descends to lower altitudes (same as indicated in the comment).

15. L278. How is the "inner boundary of the stratospheric polar vortex" defined? Also, again why does it have to be horizontal transport? I'm not saying that it isn't, but it would be good to see some explanation.

*Author response:*

This paragraph has been revised (see Comment #4) and no longer references the inner and outer boundary of the polar vortex.

16. L284. This is the only place where the term "outer boundary" is used. Similarly to "inner boundary" and "inner vortex", it is never defined. Please, provide those definitions in the methods section.

*Author response:*

This paragraph has been revised (see Comment #4) and no longer references the inner and outer boundary of the polar vortex.

17. LL343-349. This is related to my general comment #1. The way I read it Figure 8a, it shows that most of the TCO difference between Weak and Strong/Regular descent years

comes from the lower stratosphere. This makes sense because years when the vortex is relatively undisturbed due to weak wave activity tend to have stronger / more widespread heterogeneous depletion in June–September and, consequently less ozone in the lower stratosphere in October/November. In contrast, middle and upper stratospheric ozone difference, while of the same negative sign, contributes very little to the overall difference. The efficacy of MPA as a diagnostic here results from the fact that MPA is highly correlated with wave activity, not from a direct contribution of upper stratospheric transport to the size of the ozone hole. Is that a correct interpretation? At the risk of repeating myself, I think this really needs more discussion because as the paper is written one gets the impression that mesospheric transport controls the TCO (as opposed to be simply correlated with it) in October, which I don't think is correct.

**Author response:**

We hope the additional clarification from Comment #1 is helpful for interpreting this figure. We acknowledge that some of the reduction in ozone number density in the lower stratosphere in Fig. 9a (previously Fig. 8a) could indeed be from prolonged chemistry. We have altered the language describing this figure slightly to leave room for both transport and chemical effects: "*Areas with negative (blue) values in Fig. 8a indicate local deficiency of $O_3$ during a Weak descent ozone hole year, **likely due** to reduced horizontal transport as indicated by Fig. 6.*" → "*Areas with negative (blue) values in Fig. 9a indicate local deficiency of $O_3$ during a Weak descent ozone hole year, **due primarily** to reduced horizontal transport as indicated by Fig. 6.*"

18. LL350-354. Strictly speaking a metric can't "contribute" to a physical process. I interpret this statement as saying that variability of mesospheric descent (which MPA is a measure of) is a major contribution to the size of the ozone holes. But that's not the case. All this calculation shows is that MPA is a good diagnostic of the variability of October TCO, which is a fine result. But it does not demonstrate that mesospheric descent is responsible for this variability. Conversely, one could also say that TCO variability is a good metric of upper-atmosphere transport. It's a correlation vs. causality type of thing. I really suggest that this be clearly stated.

**Author response:**

The goal of the MPA is to be a good diagnostic for October TCO. As discussed earlier for Comment #1, we do not intend to infer causation. We have added additional clarification here when describing the Fig. 9b (formerly Fig. 8b):

"*The green line in Fig. 9b reflects the approximate yearly variation in TCO due to non-transport effects (e.g. chemistry or other dynamical effects not captured by the MPA), with a new standard deviation of ~13 DU.*"

19. LL367-368. Actually, since CO is averaged zonally, not within the vortex, this doesn't tell us how much time the parcel has spent in the vortex. I would appreciate a comment on this.

**Author response:**

In Fig. 2b, we find the date when the parcel first descends into the high latitude stratosphere, below 36 km. While we use a simple high-latitude CO average to track the parcel, as CO values are very low outside the vortex, this is analogous to tracking the descent of the parcel within the stratospheric polar vortex. To help illustrate the complete timeline of the parcel's descent, we have changed the x-axis units of Fig. 2b to DD-MMM. The parcel enters the stratosphere by early September in all years, which means it has spent nearly 2 months within the vortex before the MPA is calculated at the end of October.

**Other changes**

**General corrections**

1. In the process of clarifying the clustering of TCOs and the MPA, the categorical thresholds for both October ozone holes and mesospheric descent classification have been updated to reflect the mathematics of the k-means sorting process. In the first submission, cluster standard deviations were used. As the k-means process sorts years based on their proximity to the 3 cluster centroids, the categorical thresholds have now been set to the midpoints between corresponding cluster centroids. This resulted in the following minor updates in Table 1 and 2:
    a. TCO: *261 DU → 259 DU*
    b. TCO: *210 DU → 208 DU*
    c. MPA: *26.5 km → 26.9 km*
    d. MPA: *25 km → 24.6 km*
2. L. 167-168: updated the wording to accurately reflect the timeline in Fig. 1
3. Coloured symbols have been removed from all figure captions
4. L. 209: *driving from → breaking in*
5. L. 250: $O_3$ *loss* → $O_x$ *loss*
6. Throughout, where applicable: *contours → shading*
7. L. 323: *Fig. A4d → Fig. A4e*

**Wording / clarity (not covered in the above comments)**

1. L. 2: *while → as*
2. L. 54: *ozone → TCO*
3. L. 63: *"distinct datasets"* → *"daily profile time series"*
4. L 64: *"zonally averaged daily profile time series"* → *"zonal average"*
5. Table 1 caption reworded for clarity
6. L. 172: *contours → observations*

7. L. 183: *deep → deeper*
8. Fig. 1 caption edited for clarity: *"below 36 km"* and *"on the x-axis"* added
9. Items (a)-(d) on L. 185-199 have minor wording updates, for clarity
10. Fig. 2 caption minor wording update: *close → end*
11. Fig. 4 caption minor wording update: *>5 → 5*
12. Fig. 5 caption edited for clarity: *"model calculated" → "simulated"*, *"concentration" → "VMR"*, *"$O_x$ loss ≤" → "the magnitude of $O_x$ loss ≥"*
13. L. 278: *"total increase in $O_3$ alone" → "total modelled change in $O_3$"*
14. L. 279: *"transport produces a substantial increase in October ozone number density" → "transport outmatches chemical depletion of $O_3$"*
15. L. 302: *stronger/weaker → stronger (weaker)*, *Major/Non-Major → Major (Non-Major)*
16. L. 308-309: *"test this link" → "test these links to the MPA"*
17. Fig. 6 caption minor wording update: *"model simulated" → "simulated"*, *"modelled $O_3$ change alone" → "total modelled $O_3$ change"*
18. L. 311-313 and in the Fig. 7 caption: added the word *"the"* before *"start"* and *"end"*
19. L. 321: *high/low → high (low)*, *reduced/enhanced → reduced (enhanced)*
20. Table 2 caption reworded for clarity
21. L. 356-359 correction to the interpretation of the coloured marker for 2009
22. L. 363: *"with contour lines at intervals of $0.2x10^{12}$" → "in units of number density $(10^{12}$"*
23. L. 376: *tracer data → observations*
24. L. 402: *new $O_3$ → $O_3$*
25. L. 410: *new ozone → ozone*
26. L. 426: *planetary → reduced planetary*
27. L. 452: *close → end*
28. Fig. A1 and A2 captions minor wording update: *"years are presented" → "years, as labelled in each panel, are presented"*, *"months are indicated" → "months on the x-axis are indicated"*
29. Fig. A3 caption minor wording update: *"dashed" → "dash-dot"*
30. Fig. A4 caption reworded for clarity and brevity

**References added / updated**

1. L.58
2. L. 62
3. L. 80
4. L. 106
5. L. 398-399

**Spelling and grammar**

1. *ERA-5 → ERA5* throughout
2. L. 50: *sometime → sometimes*

3. L. 298: *crucial* → *a crucial*
4. L. 352: *MLA* → *MLS*
5. L. 353: *predicated* → *predicted*
6. L. 401: *the upper* → *at the upper*
7. L. 419: *complimentary* → *complementary*